# Limitations to photosynthesis by proton motive force-induced photosystem II photodamage

**Geoffry A Davis[1,2], Atsuko Kanazawa[1,3], Mark Aurel Schöttler[4], Kaori Kohzuma[1†], John E Froehlich[1], A William Rutherford[5], Mio Satoh-Cruz[1], Deepika Minhas[6], Stefanie Tietz[1], Amit Dhingra[6], David M Kramer[1,7]\***

[1]Department of Energy Plant Research Laboratory, Michigan State University, East Lansing, United States; [2]Graduate Program of Cell and Molecular Biology, Michigan State University, East Lansing, United States; [3]Department of Chemistry, Michigan State University, East Lansing, United States; [4]Max-Planck-Institut für Molekulare Pflanzenphysiologie, Potsdam-Golm, Germany; [5]Department of Life Sciences, Imperial College London, London, United Kingdom; [6]Department of Horticulture, Washington State University, Pullman, United States; [7]Department of Biochemistry and Molecular Biology, Michigan State University, East Lansing, United States

**Abstract** The thylakoid proton motive force (*pmf*) generated during photosynthesis is the essential driving force for ATP production; it is also a central regulator of light capture and electron transfer. We investigated the effects of elevated *pmf* on photosynthesis in a library of *Arabidopsis thaliana* mutants with altered rates of thylakoid lumen proton efflux, leading to a range of steady-state *pmf* extents. We observed the expected *pmf*-dependent alterations in photosynthetic regulation, but also strong effects on the rate of photosystem II (PSII) photodamage. Detailed analyses indicate this effect is related to an elevated electric field ($\Delta\psi$) component of the *pmf*, rather than lumen acidification, which *in vivo* increased PSII charge recombination rates, producing singlet oxygen and subsequent photodamage. The effects are seen even in wild type plants, especially under fluctuating illumination, suggesting that $\Delta\psi$-induced photodamage represents a previously unrecognized limiting factor for plant productivity under dynamic environmental conditions seen in the field.

*For correspondence: kramerd8@ msu.edu

**Present address:** †Graduate School of Life Sciences, Tohoku University, Sendai, Japan

## Introduction

The thylakoid proton motive force (*pmf*), the transmembrane electrochemical gradient of protons generated during the light reactions of photosynthesis, is a fundamental entity of bioenergetics, coupling light-driven electron transfer reactions to the phosphorylation of ADP via the ATP synthase (*Avenson et al., 2004*; *Kramer and Evans, 2011*). In oxygenic photosynthesis, light energy is captured by pigments in light-harvesting complexes and transferred to a subset of chlorophylls in photosystem I (PSI) and photosystem II (PSII), where it drives the extraction of electrons from water and their transfer through redox cofactors to ultimately reduce NADP+. The vectorial transfer of electrons across the membrane is tightly coupled with the generation of the *pmf*, composed of both electric field ($\Delta\psi$) and pH ($\Delta$pH) gradients.

In addition to its role in energy conservation, the *pmf* is also critical for feedback regulation of photosynthesis (*Cruz et al., 2005*). Acidification of the thylakoid lumen activates the photoprotective energy-dependent exciton quenching (q$_E$) process, which dissipates excess absorbed light energy in the photosynthetic antenna complexes by the activation of violaxanthin deepoxidase (*Demmig-*

Adams and Adams, 1992) and protonation of PsbS (Li et al., 2004). Lumen acidification also regulates the oxidation of plastoquinol by the cytochrome $b_6f$ complex, slowing electron transfer from PSII and preventing the accumulation of electrons on PSI, which can otherwise lead to photodamage (Nishio and Whitmarsh, 1993; Hope et al., 1994). In vitro work has shown that excessive lumen acidification can inactivate PSII (Krieger and Weis, 1993), decrease the stability of plastocyanin (Gross et al., 1994) and severely restrict electron flow through the cytochrome $b_6f$ complex (Kramer et al., 1999). Taken together, the in vivo and in vitro evidence of the susceptibility of photosynthetic components to acidification has led to the proposal that the extent of pmf and its partitioning into $\Delta\psi$ and $\Delta$pH components is regulated to maintain the lumen pH above about 5.8, where it can regulate photoprotection. However, under environmental stresses the regulation of photosynthesis may become overwhelmed, leading to PSII damage, or photoinhibition, from lumen over-acidification, although this has not been shown to occur in vivo (Kramer et al., 1999).

PSII photoinhibition can be a major contributor to loss of photosynthetic productivity (Raven, 2011), particularly under rapid fluctuations in environmental conditions experienced in the field (Kulheim et al., 2002). However, the mechanisms and regulation of photoinhibition remain highly debated (Keren and Krieger-Liszkay, 2011; Tyystjärvi, 2013). Though several mechanisms have been proposed for the photodamage process, it is not known which of these operate in vivo under diverse environmental conditions. In addition, there are differing views on whether the extent of photoinhibition is governed by the rate of photodamage to PSII or by regulation of PSII repair (Takahashi et al., 2007). Answering these questions is essential to understanding how plants respond to rapidly changing conditions and thus of critical importance to improving plant productivity.

The extent of the pmf can be modulated by altering the light-driven influx of protons into the lumen, i.e. by changing the rates of linear electron flow (LEF) or cyclic electron flow, or the efflux of protons through the ATP synthase (reviewed in Kramer et al., 2004). This latter mode of regulation is important for co-regulation of the light reactions with downstream metabolic processes. For example, under $CO_2$ limitations (Kanazawa and Kramer, 2002) as well as during stromal $P_i$ limited conditions (Takizawa et al., 2008) the chloroplast ATP synthase activity is strongly down-regulated, decreasing the proton conductivity ($g_H^+$) of the thylakoid, leading to buildup of pmf and activation of $q_E$. Similar decreases in $g_H^+$ and increases in pmf are seen when ATP synthase protein content is decreased (Rott et al., 2011) or in mutants with an altered ATP synthase γ-subunit (Kohzuma et al., 2012).

## Results

We took advantage of the effects of the ATP synthase on the photosynthetic proton circuit to probe how the pmf influences photoinhibition in vivo. We constructed a series of Arabidopsis thaliana mutants, which we termed minira (minimum recapitulation of ATPC2), in which we complemented a $\gamma_1$-subunit (ATPC1) T-DNA knockout line (Dal Bosco et al., 2004) with γ-coding sequences containing site-directed mutations to specifically incorporate amino acid changes around the redox regulatory cysteines present in ATPC2 into ATPC1 (Figure 1A) (Inohara et al., 1991). Each minira line is designated numerically based on amino acid position and independent transformation events; for example minira 4–1, minira 4–2, and minira 4–3 represent three independent transformation events of the same I201V mutation. The minira library was originally developed as part of an on-going 'domain swapping' approach to assess functional differences in the two ATPC paralogs in Arabidopsis, ATPC1 and ATPC2, but the minira mutants also shows a range of ATP synthase activities useful to the present work. As described below, we also confirmed key aspects of the work using previously characterized ATPC1 tobacco antisense lines (Rott et al., 2011). However, variations in ATP synthase suppression in the antisense lines between leaves and plant generations limited their experimental utility. The current mutagenesis approach was preferable as it allowed repeatable analyses of multiple photosynthetic parameters in stable, identical genetic backgrounds within each mutant line.

We assessed the effects of minira modifications on $g_H^+$ and the extent of the light-driven pmf in vivo based on the decay kinetics of the electrochromic shift (ECS), which reports changes in the thylakoid electric field (Sacksteder and Kramer, 2000). The minira lines displayed a range of ATP synthase activities (Figure 1C), from about 30–120% that of wild type (Wassilewskija-2, Ws-2), resulting

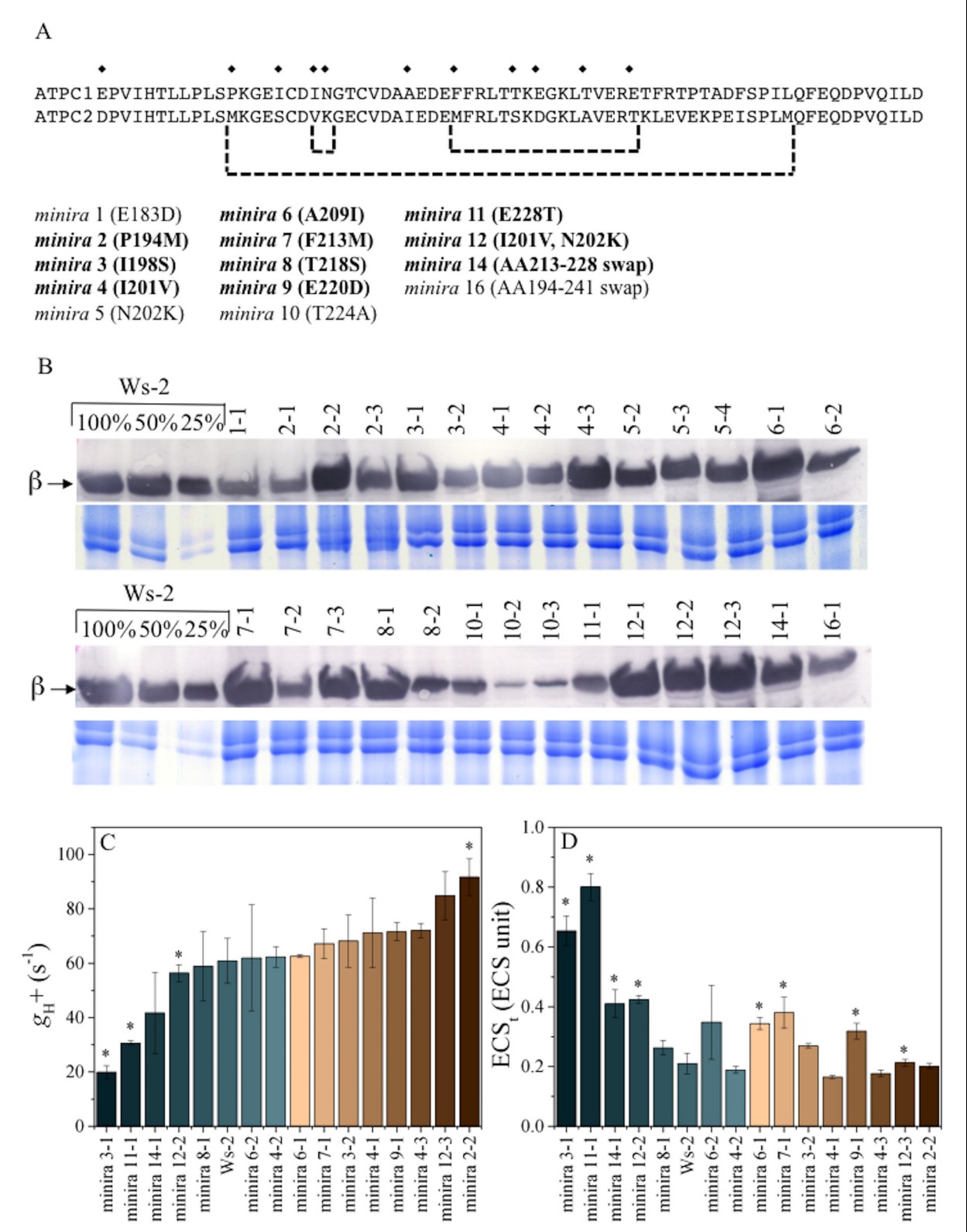

**Figure 1.** γ-subunit mutations alter photosynthetic proton efflux. Sequence alignment of Arabidopsis ATPC1 and ATPC2 regulatory region (**A**). Amino acid differences incorporated into ATPC1 to generate *minira* are indicated by symbols (♦). Amino acid numbers are based on standard spinach positions, which *minira* numeric designations are based upon position within the amino acid primary sequence. Regions where multiple changes were incorporated from ATPC2 are outlined in brackets. Bold: mutations resulting in successful transgenic plants with stable phenotypes utilized for

*Figure 1 continued on next page*

*Figure 1 continued*

experiments. Accumulation of chloroplast ATP synthase complexes was verified across resulting transformant lines (**B**). Total leaf protein was probed for chloroplast ATP synthase β-subunit compared to a titration of wild type (Ws-2) accumulation. Gels run with identical samples stained with Coomassie Brilliant Blue are shown below to ensure equal loading to the 100% wild type samples. The conductivity of the ATP synthase for protons ($g_H$+, **C**) and the light-driven *pmf* (ECS$_t$, **D**), calculated from the decay of the electrochromic shift at 100 µmol photons m$^{-2}$s$^{-1}$ actinic light (mean ± s.d, n = 3). Statistically significant differences (*p<0.05) from wild type were determined using a t-test. ECS units were defined as the deconvoluted ΔA$_{520}$ µg chlorophyll$^{-1}$ cm$^2$.

in similar variations in light-driven *pmf* (*Figure 1D*). Multiple independent transformations were utilized for the same *minira* mutation, as the $g_H^+$ changes likely reflect both intrinsic ATP synthase activity changes due to the mutations, as well as changes due to protein expression level or stability of the mutated subunit within the complex (*Figure 1B*). While some *minira* mutants display an increase in $g_H^+$ relative to the wild type and will acidify the lumen at a slower rate, others have a large decrease in total ATP synthase content. For the purpose of understanding how a high *pmf* impacts photosynthesis we have primarily focused on those mutants that modified $g_H^+$ while maintaining an ATP synthase content similar to wild type levels.

The *minira* library was then screened for photosynthetic phenotypes using whole plant chlorophyll fluorescence imaging (*Cruz et al., 2016*) over a consecutive three-day photoperiod (*Figure 2A*). Photosynthetic parameters were calculated from these images (videos are shown in *videos 1–9*) and shown as kinetic traces (*Figure 2—figure supplements 1–15*) or as log-fold changes compared to wild type (*Figure 2B–D*). Under 'standard' laboratory growth chamber lighting on day one, most *minira* lines showed relatively small differences from wild type in LEF (*Figure 2B*), q$_E$ (*Figure 2C*), and photoinhibitory quenching (q$_I$) (*Figure 2D*), with the exceptions of *minira* 3–1 and 11–1, which also showed the most severe decreases in $g_H^+$ (*Figure 1C*). Stronger photosynthetic phenotypes appeared on days two and three, implying that the decreased $g_H^+$ and *pmf* effects were enhanced by intense or fluctuating illumination.

In general, LEF decreased with decreasing $g_H^+$ (*Figure 2B*, rows are ordered by increasing $g_H$+) while q$_E$ increased (*Figure 2C*). The increases in q$_E$ were especially pronounced at higher light intensities on days two and three. These effects can be explained by slowing of proton efflux through the ATP synthase in the mutants that results in increased *pmf* for a given LEF. This is reflected in the higher lumen pH-sensitive q$_E$ response for a given LEF (*Figure 2E*), which is strikingly similar to that attributed to ATP synthase regulation in wild type plants during limitations in carbon fixation (*Kanazawa and Kramer, 2002*) or decreases in ATP synthase content (*Rott et al., 2011*). The q$_E$ sensitivities for the mutants remained similar throughout the experiments, i.e. the data for each mutant followed similar curves, implying that the ATP synthase activities were relatively constant within a particular line, consistent with a lack of light-dependent modulation of $g_H^+$ that has previously been observed (*Avenson et al., 2005*). The extents of q$_E$ did not exceed about 3.5 units in any of the lines, suggesting that either the lumen pH was restricted to a moderate acidity or that the capacity of the q$_E$ response was saturated as light intensities increased.

We also observed a strong correlation between increased q$_E$ and q$_I$, (for quantitative comparisons, see *Figure 2—figure supplement 2*), implying that decreases in ATP synthase activity in the mutants led to not only higher photoprotection but also higher rates of PSII photoinhibition. This result appears counterintuitive, in that we would expect the photoprotective q$_E$ response to prevent photoinhibition. Particularly striking was the relative loss in q$_E$ near the peak light intensities on days two and three in the most strongly affected *minira* lines (*Figure 2C*). We attribute this effect to strong accumulation of PSII photoinhibition (*Figure 2D*) leading to the loss of photosynthetic capacity (see *Figure 2B*) that limited acidification of the thylakoid lumen.

The observed increases in photoinhibition at high *pmf* could be caused by several mechanisms. It has been proposed that photoinhibition is primarily controlled by modulating the rate of PSII repair, i.e. the rate of damage is dependent solely on light intensity but repair being inhibited by stress-induced ROS production (*Nishiyama and Murata, 2014*). However, blocking PSII repair with the chloroplast translation inhibitor lincomycin (*Tyystjarvi and Aro, 1996*) revealed that, when compared to wild type or *minira* lines with wild type like $g_H^+$ (e.g. *minira* 6–1), *minira* lines with low $g_H^+$ and high *pmf* (e.g. *minira* 3–1, *Figure 1C*, *Figure 3*) had higher rates of photodamage as reflected

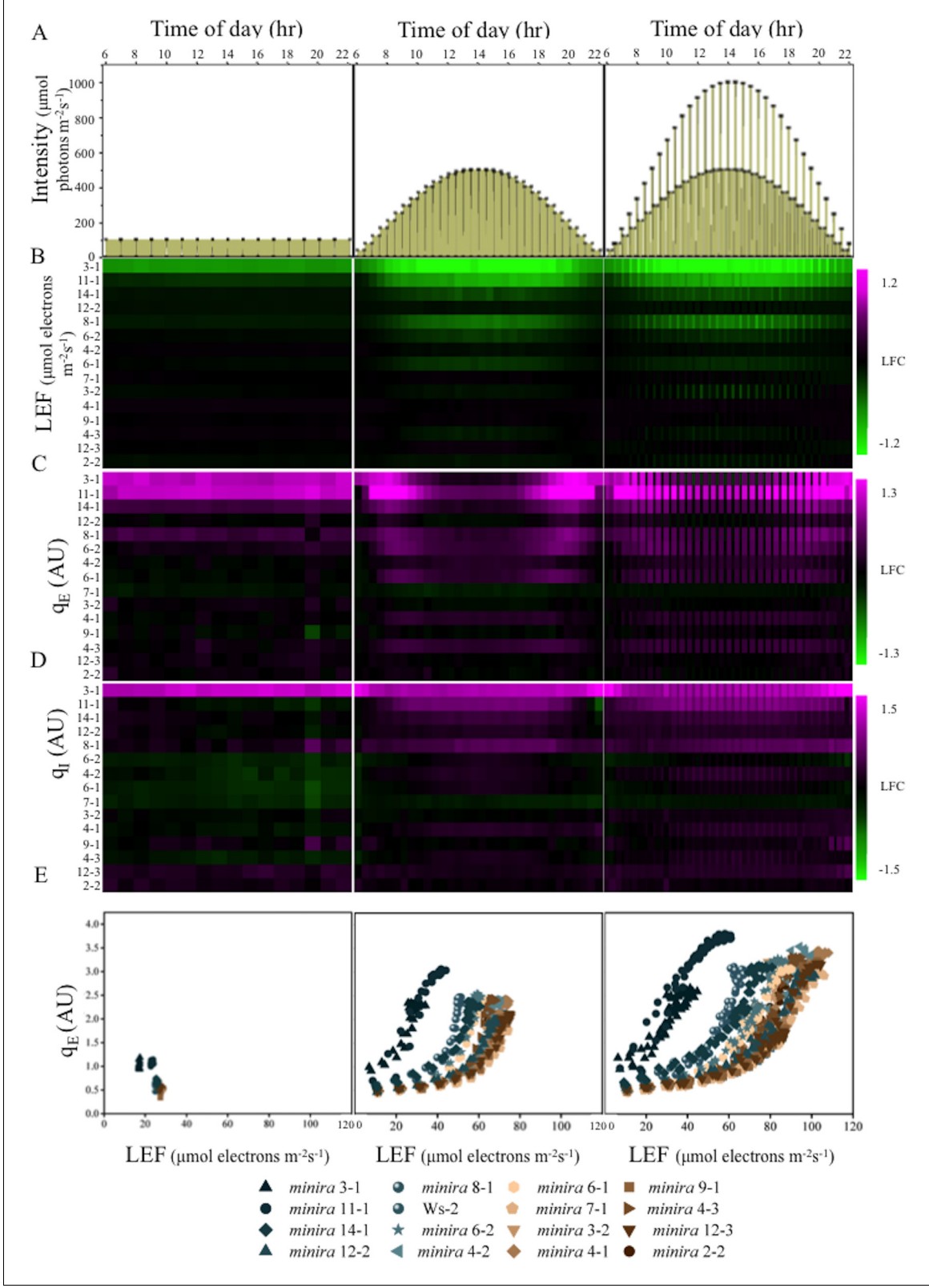

**Figure 2.** Dynamic light conditions enhance *pmf* dependent phenotypes. Whole plant fluorescent images were captured over three days under the illumination conditions displayed in Panel **A** and listed in *Supplementary file 2*. Plants were illuminated over the 16-hr photoperiod, shown as yellow filled areas representing the light intensity when present in **A**, under either a constant light intensity (day one), a sinusoidal photoperiod (day two), or a sinusoidal photoperiod interrupted by fluctuations in light intensity (day three). Square symbols in Panel **A** indicate each light intensity change. Steady-

*Figure 2 continued on next page*

*Figure 2 continued*

state fluorescence parameters were captured for each plant at the end of each light condition. Panels **B,C** and **D** represent the responses of LEF, $q_E$ and $q_I$ respectively (n ≥ 3). Data are shown as $\log_2$-fold changes compared to the wild type. Kinetic data including wild type are shown in *Figure 2— figure supplements 1–15*. The rows were sorted in order of ascending $g_H^+$ values measured as in *Figure 1C*. Panel E plots the dependence of the mean (n ≥ 3) of $q_E$ against the linear electron flow (LEF) for each time point measured for the day. For visualization purposes the error bars have been omitted from **E**.

The following figure supplements are available for figure 2:

**Figure supplement 1.** Whole plant fluorescence imaging phenotyping of *minira* 3–1 mutant.

**Figure supplement 2.** Increased pH-dependent quenching correlates with increased photoinhibitory quenching.

**Figure supplement 3.** Whole plant fluorescence imaging phenotyping of *minira* 11–1 mutant.

**Figure supplement 4.** Whole plant fluorescence imaging phenotyping of *minira* 14–1 mutant.

**Figure supplement 5.** Whole plant fluorescence imaging phenotyping of *minira* 12–2 mutant.

**Figure supplement 6.** Whole plant fluorescence imaging phenotyping of *minira* 8–1 mutant.

**Figure supplement 7.** Whole plant fluorescence imaging phenotyping of *minira* 6–2 mutant.

**Figure supplement 8.** Whole plant fluorescence imaging phenotyping of *minira* 4–2 mutant.

**Figure supplement 9.** Whole plant fluorescence imaging phenotyping of *minira* 6–1 mutant.

**Figure supplement 10.** Whole plant fluorescence imaging phenotyping of *minira* 7–1 mutant.

**Figure supplement 11.** Whole plant fluorescence imaging phenotyping of *minira* 3–2 mutant.

**Figure supplement 12.** Whole plant fluorescence imaging phenotyping of *minira* 4–1 mutant.

**Figure supplement 13.** Whole plant fluorescence imaging phenotyping of *minira* 9–1 mutant.

**Figure supplement 14.** Whole plant fluorescence imaging phenotyping of *minira* 4–3 mutant.

**Figure supplement 15.** Whole plant fluorescence imaging phenotyping of *minira* 12–3 mutant.

**Figure supplement 16.** Whole plant fluorescence imaging phenotyping of *minira* 2–2 mutant.

in both decreased maximal PSII quantum efficiency and loss of t capacity to perform charge separation in PSII (*Joliot and Delosme, 1974*; *Joliot et al., 1977*; *Bailleul et al., 2010*) and significantly decreased levels of D1 protein (*Figure 3C–E*). Thus, decreasing ATP synthase activity led to increased PSII photodamage rather than decreased rates of repair, in contradiction with strict control of photoinhibition by repair (*Takahashi et al., 2007*).

Photodamage can be induced *in vitro* by excitation of PSII centers with previously reduced primary quinone acceptor ($Q_A$) leading to the formation of the doubly-reduced $Q_AH_2$ state (*Keren and Krieger-Liszkay, 2011*). This situation might be expected if a high *pmf* slowed electron transfer through the $b_6f$ complex, resulting in the accumulation of electrons on PSII acceptors. When data at a range of light intensities are compared, the relationship between the $Q_A$ redox state ($q_L$) (*Kramer et al., 2004*) and $q_I$ is statistically significant (ANOVA, $p=2 \times 10^{-16}$) (*Figure 4*). However, both the light intensity ($p=3 \times 10^{-5}$) and $q_E$ ($p=7 \times 10^{-3}$) are significant interacting factors. At any one light intensity, $q_L$ was relatively stable, whereas $q_I$ was strongly dependent upon the mutant background and underlining *pmf* changes, indicating that while $Q_A$ reduction may be a contributing

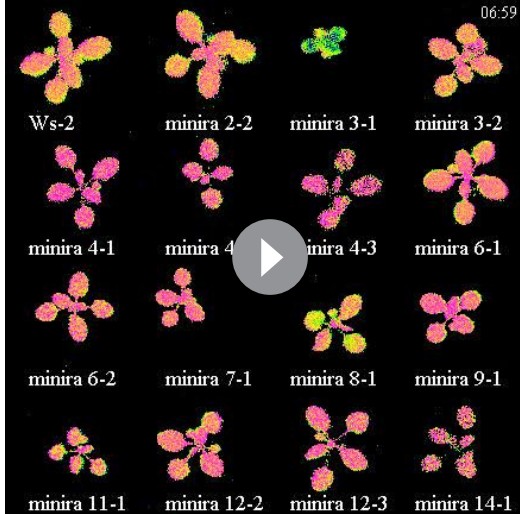

**Video 1.** (phi2_movie_labeled_day1): Whole plant PSII quantum efficiency (ΦII) during constant illumination. False-colored chlorophyll fluorescence images of whole plants obtained during constant, 100 $\mu$mol photons m$^{-2}$ s$^{-1}$ actinic illumination over a 16 hr photoperiod. Measurements and calculations were performed as described in Materials and methods. Images were false colored according to calculated values, from magenta (high) to blue (low).

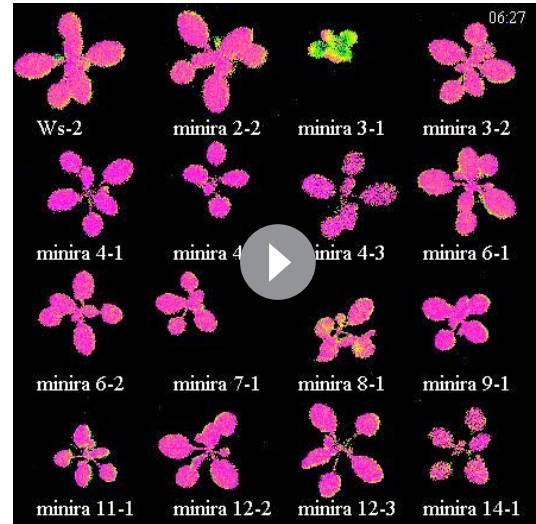

**Video 2.** (phi2_movie_labeled_day2): Whole plant PSII quantum efficiency (ΦII) during sinusoidal illumination. False-colored chlorophyll fluorescence images of whole plants obtained during sinusoidal actinic illumination over a 16 hr photoperiod. Measurements and calculations were performed as described in Materials and methods. Images were false colored according to calculated values, from magenta (high) to blue (low).

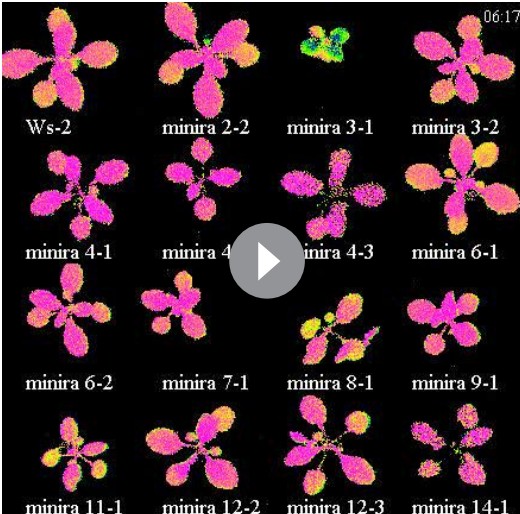

**Video 3.** (phi2_movie_labeled_day3): Whole plant PSII quantum efficiency (ΦII) during fluctuating sinusoidal illumination. False-colored chlorophyll fluorescence images of whole plants obtained during fluctuating sinusoidal actinic illumination over a 16 hr photoperiod. Measurements and calculations were performed as described in Materials and methods. Images were false colored according to calculated values, from magenta (high) to blue (low).

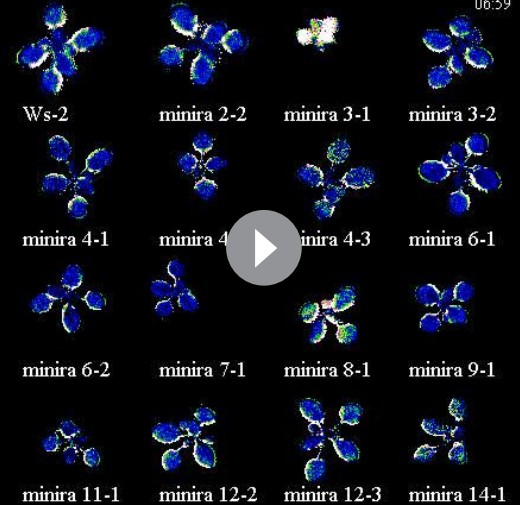

**Video 4.** qE_movie_labeled_day1): Whole plant pH-dependent quenching (qE) during constant illumination. False-colored chlorophyll fluorescence images of whole plants obtained during constant, 100 $\mu$mol photons m$^{-2}$ s$^{-1}$ actinic illumination over a 16 hr photoperiod. Measurements and calculations were performed as described in Materials and methods. Images were false colored according to calculated values, from magenta (high) to blue (low).

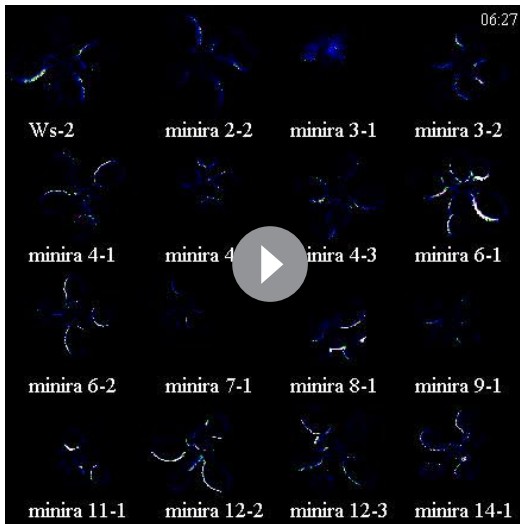

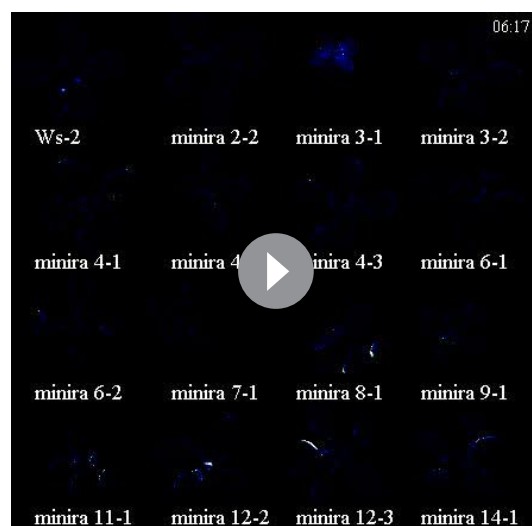

**Video 5.** (qE_movie_labeled_day2): Whole plant pH-dependent quenching ($q_E$) during sinusoidal illumination. False-colored chlorophyll fluorescence images of whole plants obtained during sinusoidal actinic actinic illumination over a 16 hr photoperiod. Measurements and calculations were performed as described in Materials and methods. Images were false colored according to calculated values, from magenta (high) to blue (low).

**Video 6.** (qE_movie_labeled_day3): Whole plant pH-dependent quenching ($q_E$) during fluctuating sinusoidal illumination. False-colored chlorophyll fluorescence images of whole plants obtained during fluctuating sinusoidal actinic illumination over a 16 hr photoperiod. Measurements and calculations were performed as described in Materials and methods. Images were false colored according to calculated values, from magenta (high) to blue (low).

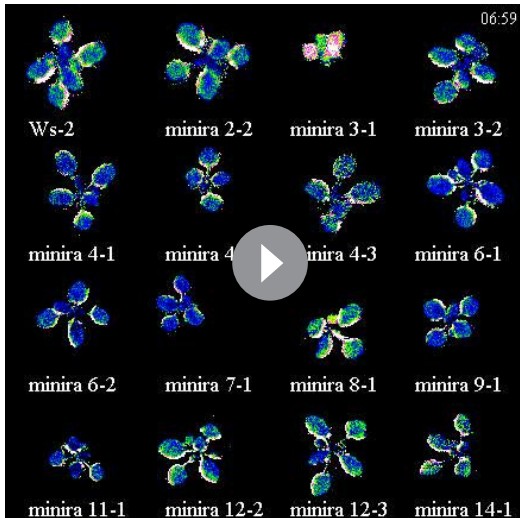

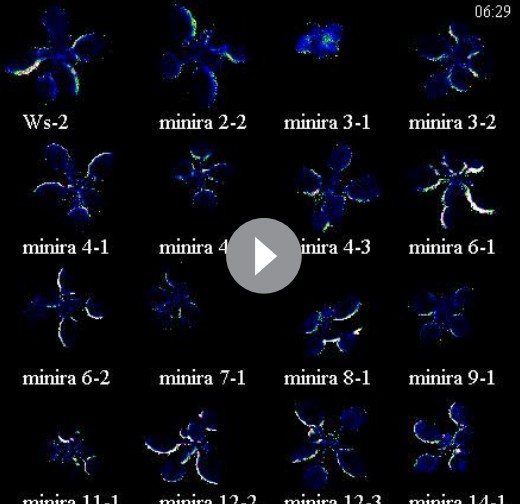

**Video 7.** (qI_movie_labeled_day1): Whole plant photoinhibitory quenching ($q_I$) during constant illumination. False-colored chlorophyll fluorescence images of whole plants obtained during constant, 100 $\mu$mol photons m$^{-2}$ s$^{-1}$ actinic illumination over a 16 hr photoperiod. Measurements and calculations were performed as described in Materials and methods. Images were false colored according to calculated values, from magenta (high) to blue (low).

**Video 8.** (qI_movie_labeled_day2): Whole plant photoinhibitory quenching ($q_I$) during sinusoidal illumination. False-colored chlorophyll fluorescence images of whole plants obtained during sinusoidal actinic actinic illumination over a 16 hr photoperiod. Measurements and calculations were performed as described in Materials and methods. Images were false colored according to calculated values, from magenta (high) to blue (low).

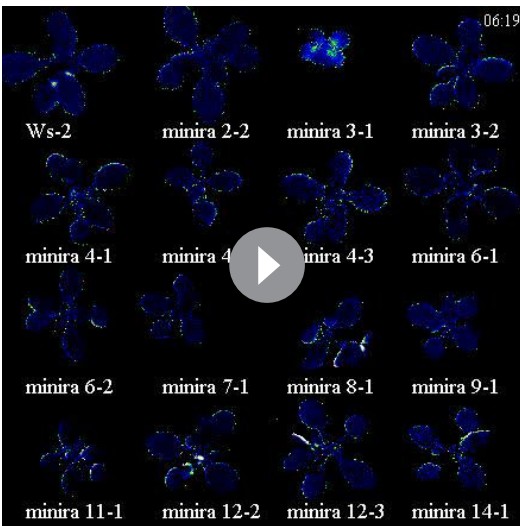

**Video 9.** (qI_movie_labeled_day3): Whole plant photoinhibitory quenching (q$_I$) during fluctuating sinusoidal illumination. False-colored chlorophyll fluorescence images of whole plants obtained during fluctuating sinusoidal actinic illumination over a 16 hr photoperiod. Measurements and calculations were performed as described in Materials and methods. Images were false colored according to calculated values, from magenta (high) to blue (low).

factor, it cannot by itself explain the observed extents of photoinhibition in the *minira* lines. On the other hand, this dependence is also consistent with an alternative model, proposed below, that involves effects on PSII recombination rates.

Changes in chlorophyll content have also been correlated with increases in PSII photoinhibition (*Pätsikkä et al., 2002*), likely due to less light being absorbed at the leaf surface and the subsequent increased light penetration into the leaf reaching more PSII centers. While the leaf chlorophyll content was altered in the *minira* mutants from wild type levels (*Supplementary file 3*), the leaf chlorophyll content does not fall below where (*Pätsikkä et al., 2002*) observed correlations between a lack of chlorophyll content and photoinhibition.

We next hypothesized that the most probable explanation for the increased photoinhibition is direct sensitization of PSII to photodamage by *pmf*. In the 'acid-damage' model (*Kramer et al., 1999*), it was proposed that excessive lumen acidification at high ΔpH (i.e. low lumen pH) could sensitize PSII centers to photodamage. To test this possibility, we compared the rates of photoinhibition with the extents of the ΔpH and Δψ components of the *pmf* by measuring the relaxation kinetics of the ECS signal (*Takizawa et al., 2007*). Surprisingly, increased extents of photoinhibition in low g$_H^+$ lines were not correlated with ΔpH, but were with Δψ (*Figure 5*), contrary to what was expected with the acid-damage model. Consistent with this result, the rates of P$_{700}^+$ reduction, which reflect the lumen pH-sensitive turnover of the cytochrome b$_6$f complex remained in a range consistent with a lumen pH above or near the pK$_a$ for b$_6$f down-regulation, i.e. above about 6.0 (*Figure 5—figure supplement 1*).

These results are consistent with an increase in the partitioning of pmf into Δψ as the total *pmf* increased (*Figure 5—figure supplement 2*), as have been observed previously (*Avenson et al., 2004*), and are likely caused by alterations in ion movements across both the thylakoid and chloroplast envelope membranes (*Avenson et al., 2004*; *Armbruster et al., 2014*; *Kunz et al., 2014*). Our results suggest that *pmf* partitioning acts to maintain a permissible lumen pH during large *pmf* increases.

We observed similar results throughout the range of *minira* mutants with low g$_H^+$ as well as with tobacco γ-subunit antisense plants (*Rott et al., 2011*) with reduced ATP synthase complexes, finding that a large increase in total *pmf* was accompanied by increased partitioning of *pmf* into Δψ (*Figure 5—figure supplement 3*), and increased rates of photodamage in the presence and absence of lincomycin (*Figure 5—figure supplement 3*). These results suggest that low g$_H^+$ or high *pmf*-related PSII damage is a more general phenomenon, which is related to excess Δψ and likely to be independent of such factors as changes in the protein or supercomplex content (see also discussion in *Rott et al., 2011*).

As described in Figure 8, we hypothesize that high Δψ accelerates photodamage by favoring recombination reactions within PSII (*de Grooth and van Gorkom, 1981*; *Diner and Joliot, 1976*; *Satoh and Katoh, 1983*; *Rappaport et al., 1999*) that lead to the formation of chlorophyll triplet states that in turn generate $^1$O$_2$ (*Johnson et al., 1995*).

To test this model, we studied the relationship between elevated Δψ and PSII charge recombination in isolated spinach thylakoids (*Figure 6*), which unlike Arabidopsis can be isolated as highly intact chloroplasts and tightly coupled thylakoids. Consistent with our model, we found that

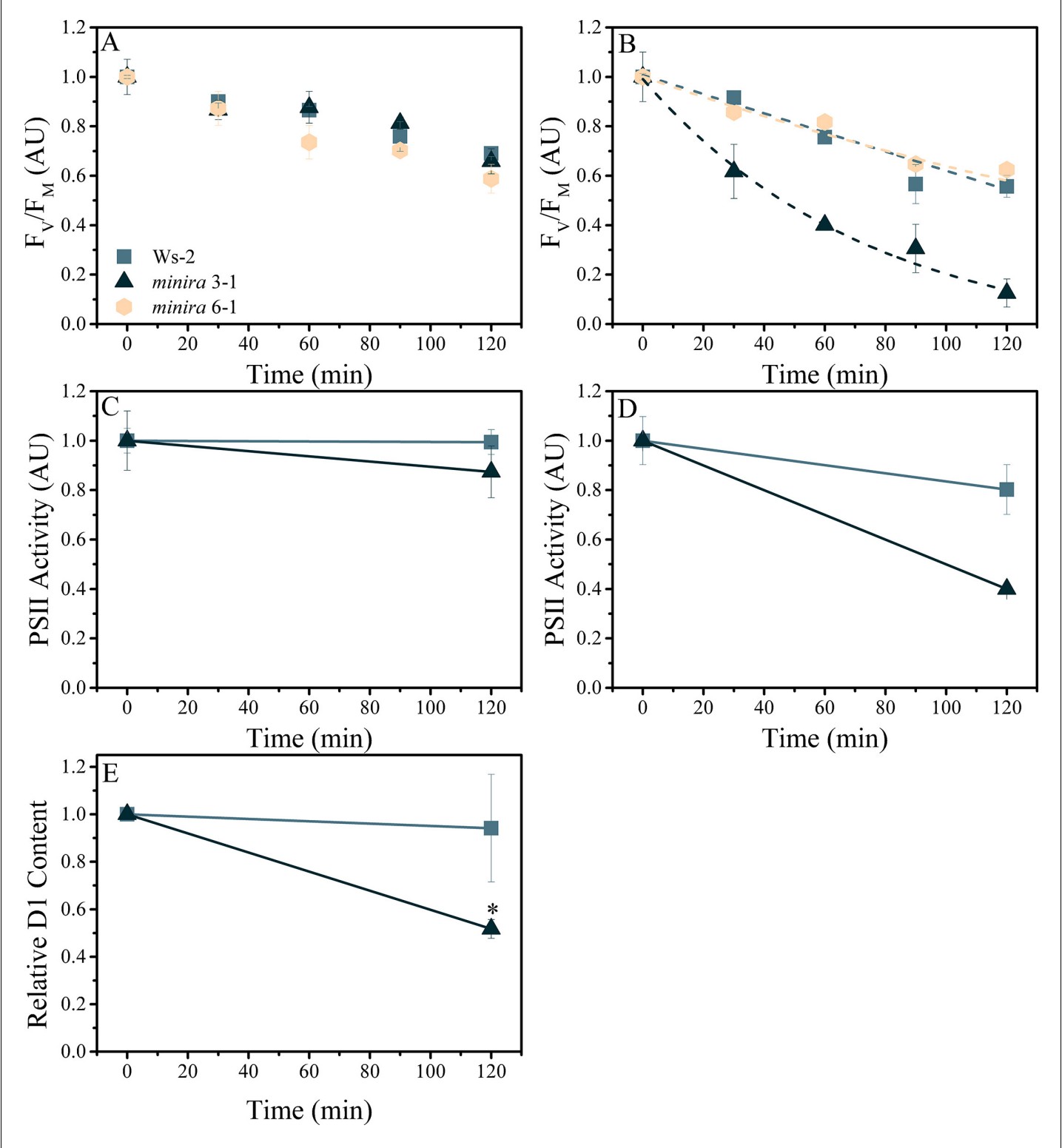

**Figure 3.** Elevated *pmf* leads to PSII photodamage. Detached leaves were infiltrated with either water (**A**,**C**) or a 3 mM solution of lincomycin (**B**,**D**) and treated with 1000 µmol photons m$^{-2}$s$^{-1}$ red light for the times indicated. Following dark adaptation, $F_V/F_M$ values were obtained (**A**,**B**) for treated leaves of wild type (square, initial $F_V/F_M$ 0.75 ± 0.006 and 0.76 ± 0.012 for water and lincomycin, respectively), *minira* 3–1 (triangle, initial $F_V/F_M$ 0.60 ± 0.07 and 0.54 ± 0.10 for water and lincomycin, respectively), and *minira* 6–1 (hexagon, initial $F_V/F_M$ 0.72 ± 0.004 and 0.73 ± 0.01 for water and lincomycin, respectively) (mean ± s.d., n ≥ 3). Data were normalized to the initial dark-adapted $F_V/F_M$ values to remove intrinsic differences between the three lines. In panel B, dashed lines represent the best fit curves for a single exponential decay. The ability of photoinhibited leaves to perform PSII

*Figure 3 continued on next page*

*Figure 3 continued*

charge separation was determined in Ws-2 and *minira* 3–1 by measuring the ECS absorbance changes following two consecutive single-turnover saturating flashes in the presence of DCMU (**C,D**). Leaves infiltrated with water (**C**, initial amplitudes of $8.88 \times 10^{-4} \pm 9.0 \times 10^{-5}$ and $9.59 \times 10^{-5} \pm 3.6 \times 10^{-5}$ for wild type and *minira* 3–1, respectively) or 3 mM lincomycin (**D**, initial amplitudes of $8.59 \times 10^{-4} \pm 1.2 \times 10^{-4}$ and $1.63 \times 10^{-4} \pm 3.1 \times 10^{-5}$ for wild type and *minira* 3–1, respectively) were infiltrated with DCMU following the indicated light treatment time and dark adaptation. PSII activity was determined by subtracting the ECS amplitude induced by the second flash from the ECS amplitude induced by the first flash (mean ± s.d., n ≥ 4). Loss of the PSII reaction center D1 protein over the time course of illumination in lincomycin treated wild type and *minira* 3–1 leaves (**E**). Leaves treated as in panel B were analyzed by western blot (n = 4) using an α-PsbA antibody and the 32 kDa band was quantified. Band intensities were normalized to the time zero point for each genotype within a single blot to control for differences in development intensities. Statistically significant differences (*p<0.05) from wild type were determined using a t-test.

elevated $\Delta\psi$, produced by artificial decyl-ubiquinol mediated cyclic electron flow through PSI, increased the rate of charge recombination from the $S_2Q_A^-$ state compared to samples treated with gramicidin to dissipate $\Delta\psi$ (***Figure 6A***). This recombination reaction can also occur *in vivo* during normal turnover when the $S_2Q_A^-$ state is formed given that the equilibrium constant for sharing of electrons between $Q_A$ and $Q_B$ is small (***Robinson and Crofts, 1983***). The extent of $\Delta\psi$ generated in

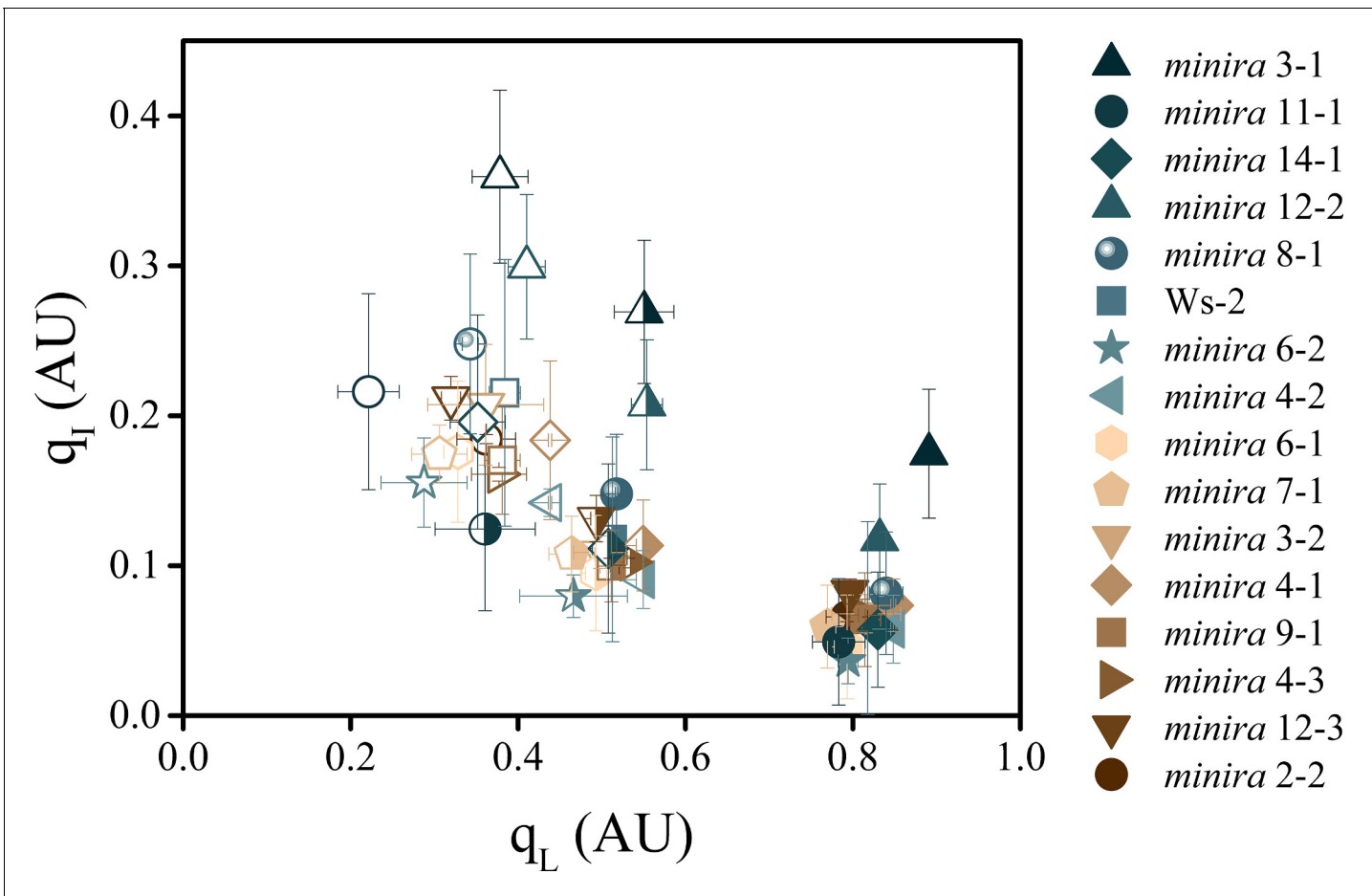

**Figure 4.** The dependence of photoinhibition on the redox state of $Q_A$. The redox state of the primary electron acceptor $Q_A$ was assayed using the $q_L$ fluorescence parameter concurrently with photoinhibitory quenching $q_I$ at 100 (solid symbols), 300 (half filled symbols), and 500 µmol photons $m^{-2}s^{-1}$ (open symbols, mean ± s.d., n = 3). Plants were exposed to at least 10 min of actinic illumination prior to $q_L$ measurement, and $q_I$ measured after 10 min of dark relaxation. While the extent of $q_I$ varies between plants, the relative redox state of $Q_A$ remains similar between all plants within each actinic light intensity.

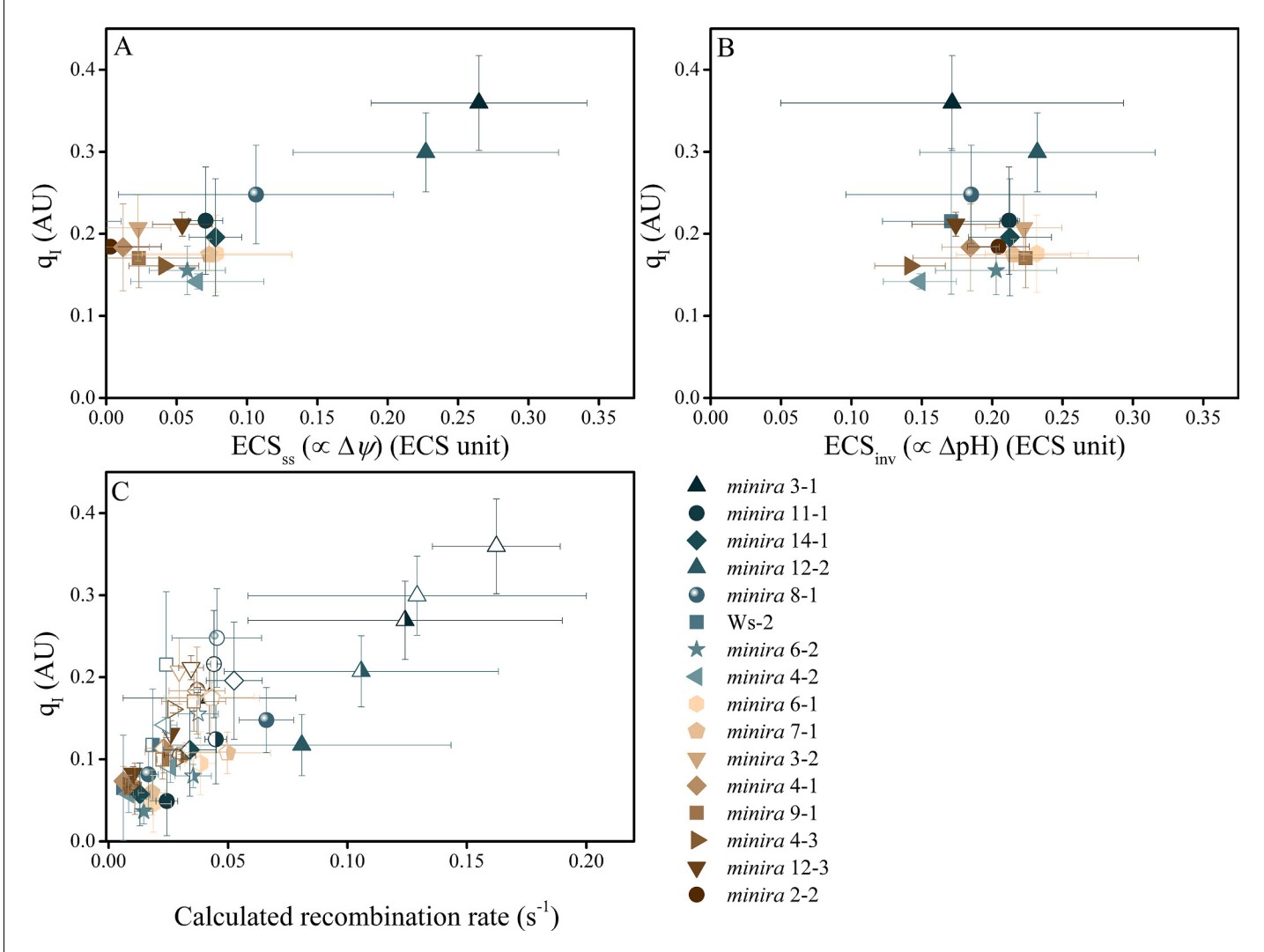

**Figure 5.** Photoinhibition is strongly correlated with Δψ but not ΔpH in *minira* lines. Photoinhibition, estimated by the $q_I$ fluorescence parameter, is plotted against either the ΔpH or Δψ components of *pmf*, estimated by the $ECS_{ss}$ (A) and $ECS_{inv}$ (B) parameters, as described in Materials and methods. Measurements shown were taken during exposure to 500 μmol photons $m^{-2}s^{-1}$ actinic light (mean ± s.d., n = 3). Two-way analysis of variance (ANOVA) of all combined data, 15 *minira* lines and wild type, showed a stronger correlation between $q_I$ and Δψ ($F$ = 9.5, p=0.003) than ΔpH ($F$ = 4.05, p=0.05). This correlation is also seen with the expected pH-dependent alterations of $P_{700}^+$ reduction for the observed partitioning differences from wild type (*Figure 5—figure supplement 1*) and an increase in the fraction of total *pmf* stored as Δψ at the expense of ΔpH over multiple light intensities (*Figure 5—figure supplement 2*). Increased storage of *pmf* as Δψ is also observed in tobacco ATP synthase knock-down plants (*Figure 5—figure supplement 3*). ECS units were defined as the deconvoluted $ΔA_{520}$ μg chlorophyll$^{-1}$ cm$^2$. The influence of Δψ on the rate of PSII recombination was estimated based on the change in the equilibrium constant for the sharing of electrons between pheophytin and $Q_A$ (described in Materials and methods) (C). The influence of Δψ on the calculated recombination rate taking into account the fraction of reduced $Q_A$ using the equations described in Materials and methods and described in the main text. Data were obtained at 100 (solid symbols), 300 (half filled symbols), and 500 μmol photons $m^{-2}s^{-1}$ (open symbols) (mean ± s.d., n = 3).

The following figure supplements are available for figure 5:

**Figure supplement 1.** Reduction kinetics of $P_{700}$+.

**Figure supplement 2.** The electric field component of the *pmf* dominates under high *pmf* conditions.

**Figure supplement 3.** Tobacco *ATPC1* antisense knockdown increase Δψ partitioning under high *pmf* conditions.

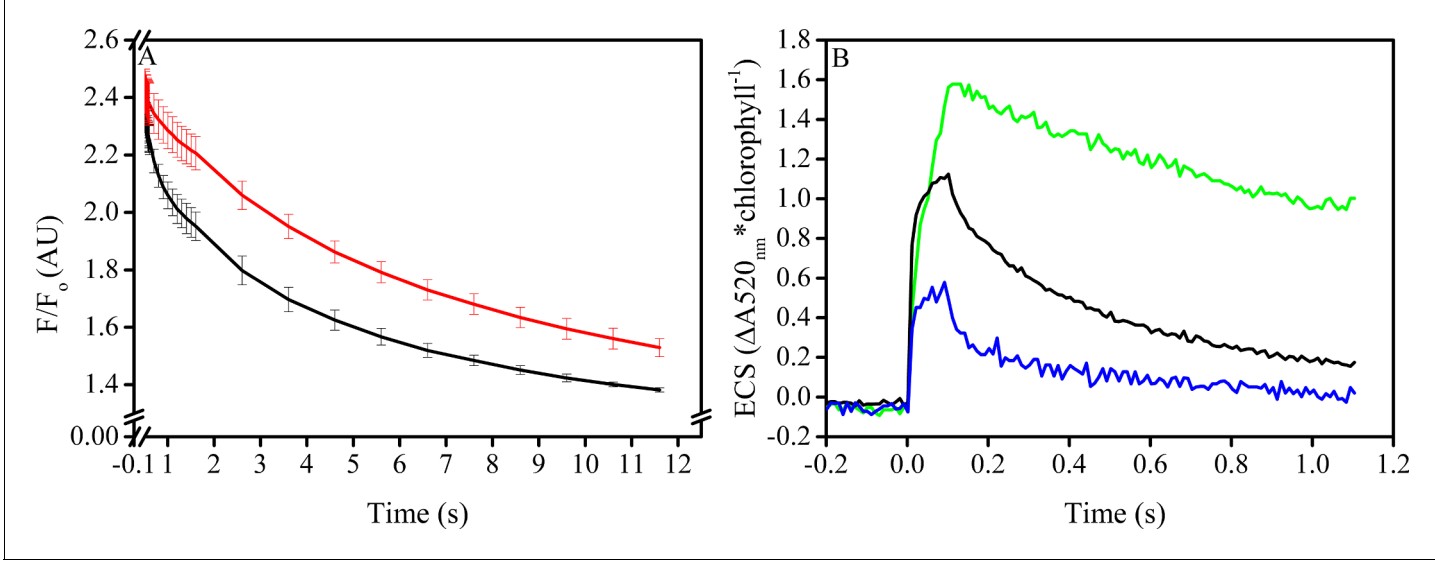

**Figure 6.** *In vitro* manipulation of the *pmf* Δψ alters PSII $S_2Q_A^-$ recombination rates. Isolated spinach thylakoids in the presence of 5 µM spinach ferredoxin and 10 µm sodium ascorbate were treated with 3–3,4-dichlorophenyl 1,1-dimethylurea (DCMU) to block PSII forward electron transfer, and a trans-thylakoid *pmf* generated utilizing decyl-ubiquinol mediated PSI cyclic electron transfer (**A**). Recombination from the $S_2Q_A^-$ state was probed by observing the decrease in the high fluorescence state associated with $Q_A^-$ following a short (100 ms) actinic flash to dark-adapted thylakoids (black line). Depletion of the *pmf*, which under these conditions is stored almost exclusively as Δψ, in the presence of 25 µM gramicidin (red line) resulted in an approximate 5-fold increase in the initial rate of decay when Δψ was largest, and an overall 2-fold increase in the lifetime of the high fluorescence state. The extent of Δψ generated by the 100 ms light (**B**), estimated by the ECS signal measured at 520 nm and normalized to chlorophyll content. Thylakoids were assayed in the absence of inhibitors (green line), in the presence of DCMU (blue line), and in the presence of both DCMU and 50 µM decyl-ubiquinol (black line) to generate *pmf* through PSI turnover, corresponding to the condition used in panel A.

these experiments was similar to that observed in *minira* leaves under photoinhibitory conditions (*Figure 6B*), suggesting similar increases in recombination rates should occur *in vivo*.

We next tested the predicted connection between elevated Δψ and singlet oxygen ($^1O_2$) generation (*Figure 7D*). In wild type leaves, moderate illumination (30 min of 300 µmol photons $m^{-2}s^{-1}$) resulted in no detectable light dependent changes in $^1O_2$ (*Figure 7D*) using Singlet Oxygen Sensor Green (SOSG) dye fluorescence (*Dall'Osto et al., 2012*; *Ramel et al., 2013*; *Shumbe et al., 2016*). In contrast, the low $g_H^+$ *minira* 3–1 line showed a strong induction of SOSG fluorescence within the first 10 min of illumination, which saturated by about 30 min. Infiltration of leaves with valinomycin, a potassium ionophore that decreases the Δψ component of *pmf* (*Satoh and Katoh, 1983*), partially inhibited the rise in in SOSG fluorescence (*Figure 7—figure supplement 1*). While care must be taken making quantitative estimates of $^1O_2$ from SOSG fluorescence (*Koh and Fluhr, 2016*), within the limits of these experiments the Δψ-dependence of the SOSG fluorescence increases strongly support a role for Δψ-induced $^1O_2$ production from photosynthesis.

Singlet oxygen is produced by the interaction of $O_2$ with triplet excited states of pigments, most likely from the $^3P_{680}$ chlorophyll within the PSII reaction centers generated by recombination reactions (*Krieger-Liszkay et al., 2008*; *Telfer, 2014*), as further discussed below. It is unlikely that such triplets could be generated in the bulk light harvesting pigments because high NPQ in the mutants will decrease the lifetime of antenna excited states, and chlorophyll triplets generated in light harvesting complexes are efficiently quenched by carotenoids (*Telfer et al., 2008*). We thus propose that elevated Δψ induces $^1O_2$ production by accelerating PSII recombination in low $g_H^+$ mutants when *pmf* is large. Keren et al. (*Keren et al., 1997*) suggested that recombination-induced triplet formation could explain the photoinhibitory effects of very low light, when PSII charge recombination is preferred over the forward electron transfer reactions. Our work implies that this type of phenomenon is greatly accelerated by high Δψ, potentially making it relevant to photosynthesis under growth light conditions.

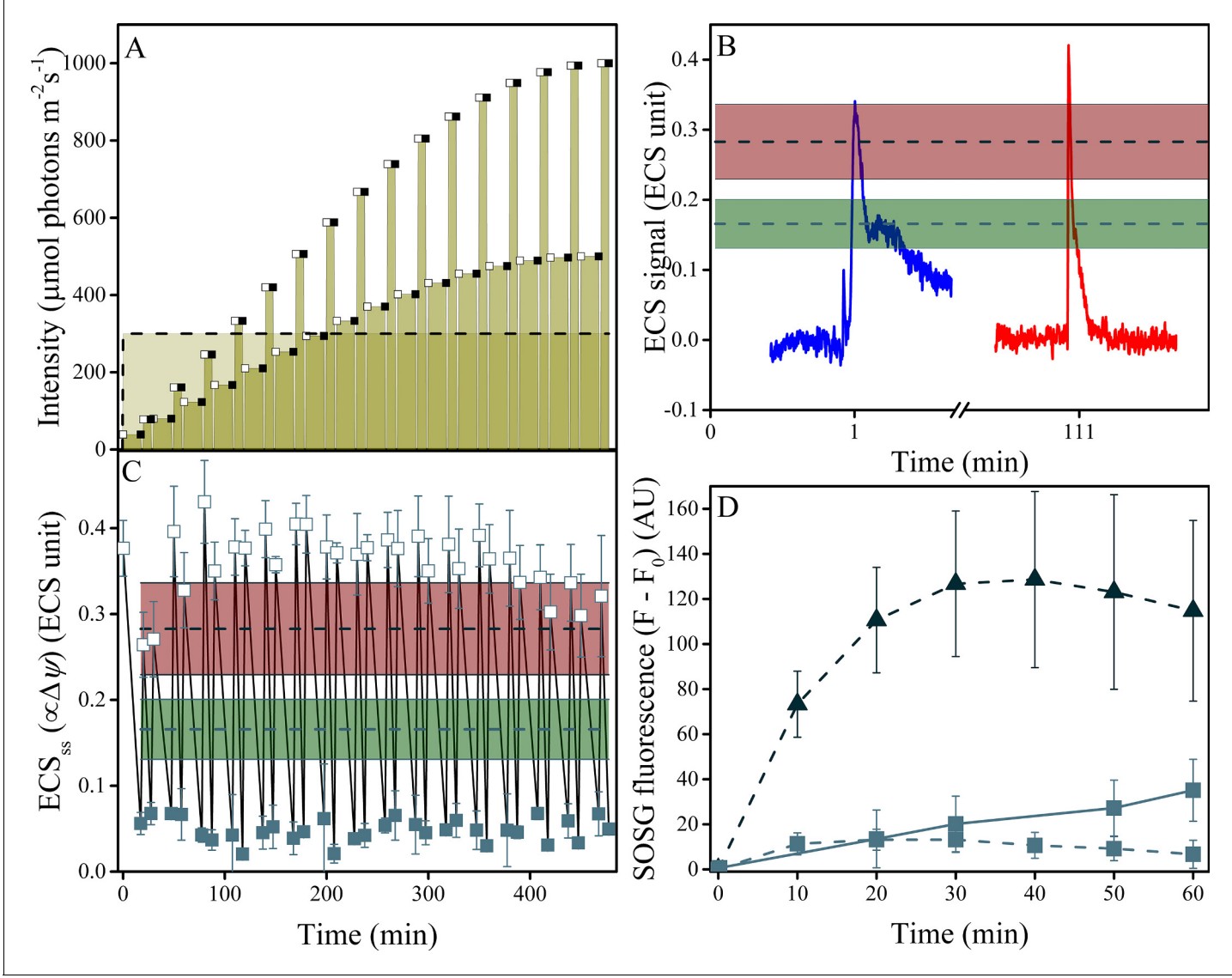

**Figure 7.** Induction of $\Delta\psi$ and $^1O_2$ production under fluctuating light in wild type plants. (**A**) Illumination conditions and measurement points used in the experiments. Fluctuating light conditions (replicating *Figure 2* day three) are shown as connected points, with open squares representing measurements obtained 10 s after the light transition and closed squares the end of steady-state illumination. Constant illumination of 300 μmol m$^{-2}$s$^{-1}$ is represented as a dotted line. (**B**) Representative traces of the light-fluctuation induced ECS signals resulting in transient ECS 'spikes' are shown for the first fluctuation (dark to 39 μmol m$^{-2}$s$^{-1}$, blue) and the fluctuation from 167 to 333 μmol m$^{-2}$s$^{-1}$, red). A full set of ECS kinetic 'spikes' following increased light fluctuations can be found in *Figure 7—figure supplement 2*. The extents of light-induced $\Delta\psi$ in wild type (green line and shaded box, indicating mean ± s.d) and *minira 3–1* (red lines and box, indicating mean ± s.d) at 300 μmol m$^{-2}$s$^{-1}$ are shown for comparison. (**C**) The extents of light-induced $\Delta\psi$, estimated using the ECS$_{ss}$ parameter over the time-course of the fluctuating light experiment, compared to those obtained under continuous illumination in wild type and *minira 3–1* (green and red lines and boxes, respectively, as in Panel B). Open and closed squares correspond to the ECS$_{ss}$ measurements taken at the timing designated in panel A. (**D**) Time-course of SOSG fluorescence changes for wild type (squares) and *minira 3–1* (triangles) during exposure to constant 300 μmol m$^{-2}$s$^{-1}$ (dotted lines)and wild type leaves under the first hour of fluctuating light (solid line). A decrease in SOSG fluorescence occurs when $\Delta\psi$ is collapsed with the addition of the ionophore valinomycin (*Figure 7—figure supplement 1*). All data in A, B, and C represent mean (n ≥ 3) ± s.d. ECS units were defined as the deconvoluted $\Delta A_{520}$ μg chlorophyll$^{-1}$ cm$^2$.

The following figure supplements are available for figure 7:

**Figure supplement 1.** Uncoupling $\Delta\psi$ decreases SOSG fluorescence in *minira* 3–1.

**Figure supplement 2.** Fluctuations in light intensity result in transient ECS spikes.

The effects of high $\Delta\psi$ would be expected to alter the rates of PSII recombination through $P^+Pheo^-$ (where $P^+$ is the oxidized primary chlorophyll donor and Pheo the D1 subunit pheophytin) in an increasing dependence upon the fraction of $Q_A^-$, consistent with the observed correlation between $q_I$ and $q_L$ as the light intensity increased (*Figure 4*). To test this relationship, we estimated the recombination rates from $S_2Q_A^-$ through the $P^+Pheo^-$ pathway, considering $Q_A$ redox state (estimated by $1$-$q_L$) and the expected impact of $\Delta\psi$ on the equilibrium constant for sharing electrons between Pheo and $Q_A$ (based on $ECS_{ss}$ and the position of $Q_A$ in the structure relative to the membrane dielectric). The basis of this estimate is described in more detail in Materials and methods. As shown in *Figure 5C*, we see a positive correlation between $q_I$ and estimated recombination through $P^+Pheo^-$ over both mutant variants and light intensities, indicating that the combined effects of $\Delta\psi$ and $Q_A$ redox state can explain a large fraction of the observed extents of photoinhibition. While it is likely that multiple mechanisms of photoinhibition exist, which may also explain some of the $q_I$ variation, overall the greatest impact upon photoinhibition under these conditions can be explained by $\Delta\psi$-mediated changes in PSII electron recombination.

The obvious questions are: does $\Delta\psi$-induced photoinhibition occur in wild type plants and if so under what conditions? Photosynthesis is known to be particularly sensitive to rapid fluctuations in light intensity (*Kulheim et al., 2002*), at least some of this sensitivity is associated with photoinhibition of PSI, especially in cyanobacteria (*Allahverdiyeva et al., 2015*). However, such fluctuations should also result in large transient changes in $\Delta\psi$, as the thylakoid membrane has a low electrical capacitance and low permeability to counter-ions, while the lumen and stroma have high proton buffering capacity (*Cruz et al., 2001*). The slow onset of $q_E$ and other down-regulatory processes in photosynthesis should exacerbate these effects and allow for large fluxes of electrons when light levels are rapidly increased, resulting in large, transient $\Delta\psi$ 'pulses'.

We therefore hypothesized that $\Delta\psi$-induced photodamage may contribute to the increased photodamage seen under fluctuating light. To test this possibility, we measured light-driven *pmf* ($\Delta\psi$ and $\Delta pH$) and $^1O_2$ generation in wild type Arabidopsis under fluctuating light conditions (*Figure 7A*, replicating the first 8 hr of *Figure 2A* day three). The initial dark-to-light transition resulted in an immediate, transient 'spike' in $\Delta\psi$, even though the light intensity was low ($39$ μmol photons m$^{-2}$s$^{-1}$, *Figure 7B*). The spike was transient, and decreased to steady-state levels within tens of seconds, as shown in the blue trace in *Figure 7B*. Spikes in $\Delta\psi$ of similar amplitudes were also seen upon each increase in light at the onset of each fluctuation, though the recovery kinetics tended to be more rapid than those seen at the first dark-light transient. An example of these transients, taken at the transition between $167$ and $333$ μmol photons m$^{-2}$ s$^{-1}$ is shown in (*Figure 7B*). A more complete set of transient kinetics, over the entire course of the experiment is presented in *Figure 7—figure supplement 2*. The amplitudes of these $\Delta\psi$ transients were similar to or larger than those seen in the *minira* 3–1 line under constant $300$ μmol photons m$^{-2}$ s$^{-1}$ light, which also induced $^1O_2$ generation (see red horizontal bars in *Figure 7B*). These spikes reflect increases in $\Delta\psi$ above that already produced by steady-state photosynthesis, so the true extent of $\Delta\psi$ is likely considerably higher, and based on estimates of the calibration of the ECS signal likely range between 150–260 mV (see *Figure 7—figure supplement 2*).

By contrast, wild type plants under constant $300$ μmol photons m$^{-2}$ s$^{-1}$ produced much lower $\Delta\psi$ extents (*Figure 7B* green bars) and had no detectible $^1O_2$ generation (*Figure 7D*). These results suggest that even low amplitude light fluctuations are capable of inducing $\Delta\psi$ large enough to produce $^1O_2$ and PSII photodamage. Supporting this interpretation, wild type leaves under these fluctuating light conditions produced substantial amounts of $^1O_2$ during the first hour of fluctuating conditions compared to higher intensity, but constant illumination (*Figure 7D*).

The above results lead us to conclude that fluctuating light likely induces strong effects through $\Delta\psi$-induced recombination reactions in PSII (see below for considerations of potential contributions to PSI). In addition to direct damage to the enzymes of photosynthesis, $^1O_2$ can also activate plant light stress-related gene expression and programmed cell death (*Zhang et al., 2014*), suggesting a possible physiological linkage between *pmf*-enhanced recombination and plant regulatory pathways that may result in long-term acclimation to fluctuating light.

At a mechanistic level, we propose that imposing a large $\Delta\psi$ across the PSII complex will decrease the standard free energy gap between the vectorial electron transfer steps and thus the back-reaction will be accelerated in competition with forward (energy-storing) reactions (*de Grooth and van*

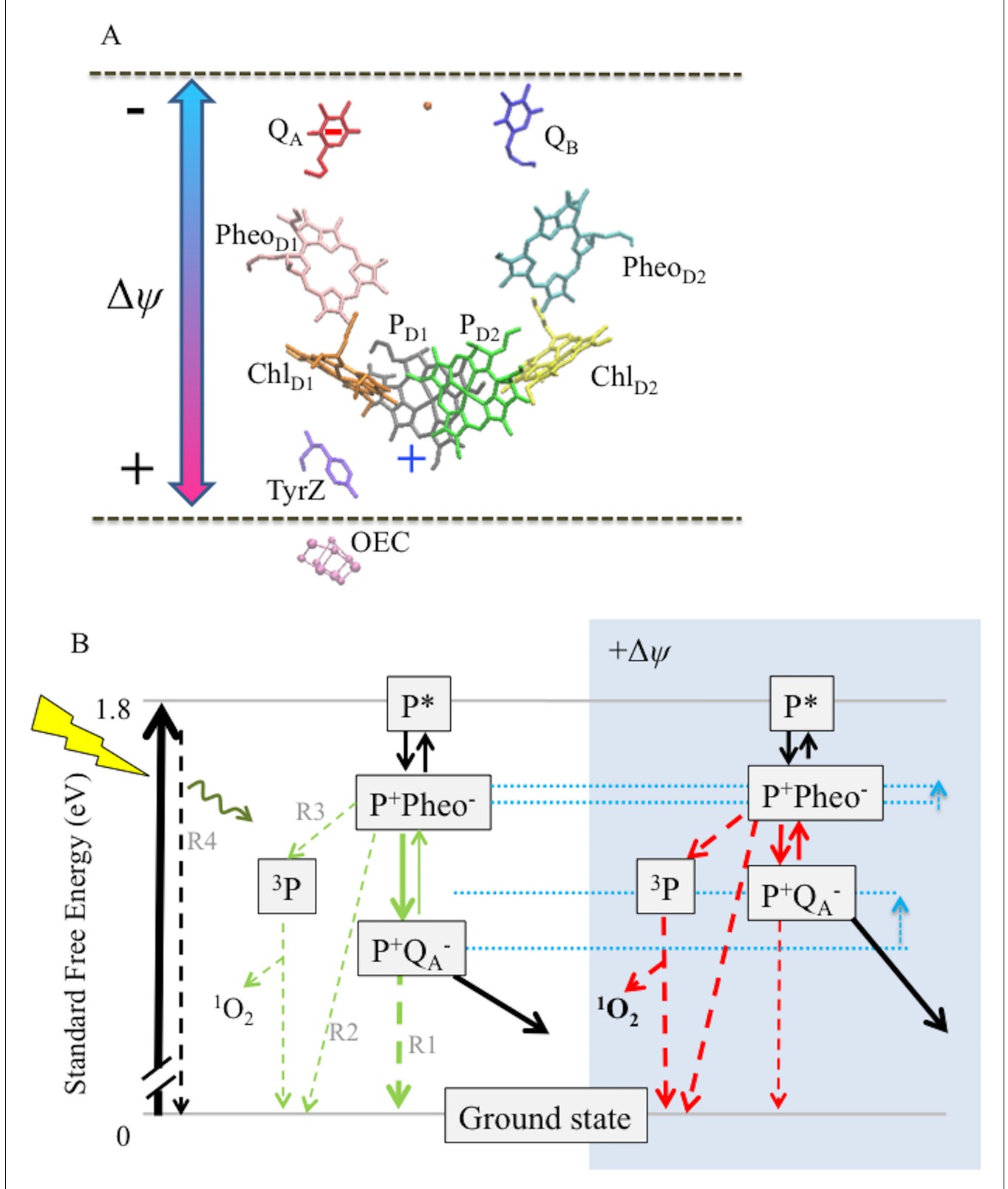

**Figure 8.** Schemes for the trans-thylakoid $\Delta\psi$-induced acceleration of recombination reactions in PSII and subsequent production of $^1O_2$. (**A**) The relative positions of PSII electron transfer cofactors with respect to the electric field (double-headed arrow) imposed across the thylakoid membrane (dotted lines). The red and blue arrows indicate the $\Delta\psi$-induced changes in the equilibrium constant for the sharing of electrons (−) between $Q_A$ and Pheo, and electron holes (+) among $P^+$ and the oxygen evolving complex. Excitation of PSII by light leads to formation of excited chlorophyll states

*Figure 8 continued on next page*

*Figure 8 continued*

(P*), the excitation is shared over the 4 chlorophylls and the 2 pheophytins. Charge separation occurs between more than one pair of pigments, so at short times the situation is not well defined, but $Chl_{D1}^+Pheo_{D1}^-$ appears to be the dominant radical pair. Secondary electron transfer events occur forming $P_{D1}^+Pheo_{D1}^-$, the second radical pair, which is present in nearly all centers. This radical pair is stabilized by electron transfer from $Pheo^-$ to $Q_A$ forming $P_{D1}^+Q_A^-$. This radical pair is further stabilized by electron transfer from D1Tyr161 (TyrZ) forming a neutral tyrosyl radical, which oxidizes the Mn cluster of the oxygen evolving complex to form the state $S_{n+1}Q_A^-$. Finally, $Q_A^-$ reduces $Q_B$ to form $S_{n+1}Q_B^-$. Upon a second PSII turnover the double reduced and protonated $Q_B$ plastohydroquinone becomes protonated and is exchanged with an oxidized plastoquinone from the membrane pool (black arrows). The illustration was based on crystal structure 3WU2 (*Umena et al., 2011*). (B) The charge separation states described above are unstable and recombination competes with the energy-storing reactions. When $P_{D1}^+$ is present, recombination reactions can occur by several pathways as indicated by the dashed lines: (1) direct electron transfer from $Q_A^-$ to $P^+$ (R1); (2) by the back reaction to form the $P^+Pheo^-$ state, which can then recombine directly (R2) or, (3) when the $P^+Pheo^-$ radical pair is present as a triplet state, $^3[P^+Pheo^-]$, the dominant state when formed by the back-reaction, $^3[P^+Pheo^-]$ charge recombination forms $^3P$ (R3), a long lived chlorophyll triplet that can easily interact with $O_2$ to form $^1O_2$; (4) complete reversal of electron transfer can also occur, repopulating $P^*$ (R4), which can return to the ground state by emitting fluorescence (luminescence) or heat. Route 3, the triplet generating pathway, is the dominant recombination route in fully functional PSII. For simplicity the $^3[P^+Phe^-]$ is not distinguised from the singlet form in this scheme. A $\Delta\psi$ across the membrane should destabilize $P^+Q_A^-$ relative to the other states (see dotted blue lines), affecting the rates of reactions indicated in the green versus red. A $\Delta\psi$ across the membrane should also destabilize $P^+Pheo^-$, but because of the smaller dielectric span across the membrane, to a lesser extent than $P^+Q_A^-$. Thus, the buildup of $\Delta\psi$ should shift the equilibrium constant for sharing electrons between $Q_A^-$ and Pheo, favoring the formation of $Pheo^-$ and thus increasing the rate of recombination through R3 (as well as the R2 and R4), resulting in increased production of $^3P$ and $^1O_2$. Destabilization of $P^+Q_A^-$ will also increase the driving force for $P^+Q_A^-$ recombination via R1, however this recombination is already driven by 1.4eV and it is thus likely to be in the Marcus inverted region. Thus increasing the driving force will slow recombination by this route.

*Gorkom, 1981*; *Vos et al., 1991*; *Johnson et al., 1995*) (*Figure 8*). It is known that decreasing the energy gap between Pheo and $Q_A$ favors recombination from $P^+Q_A^-$ via $Pheo^-$ (*de Grooth and van Gorkom, 1981*; *Johnson et al., 1995*) rather than directly from $Q_A^-$ to $P^+$ (*Johnson et al., 1995*; *Krieger-Liszkay and Rutherford, 1998*; *Rutherford et al., 2012*). The observed increased recombination rates *in vitro* (*Figure 6*) when $\Delta\psi$ is present, combined with $^1O_2$ production under high $\Delta\psi$ conditions *in vivo* suggest that the fraction of electrons recombining to $P^+$ through a $Pheo^-$ intermediate is greatly increased, as the recombination pathway via the $Pheo^-$ has a high yield for formation of the triplet state of P ($^3P$) that in turn can interact with $O_2$ to produce $^1O_2$ (*Johnson et al., 1995*; *Keren and Krieger-Liszkay, 2011*; *Rutherford et al., 2012*).

## Discussion

This proposed mechanism of $\Delta\psi$-mediated photoinhibition has broad implications for the energy limitations of photosynthesis. Although the two components of *pmf* are energetically equivalent for driving ATP synthesis (*Hangarter and Good, 1982*), they have distinct effects on the regulation of photosynthesis (*Kramer et al., 1999*; *2004*; *Finazzi et al., 2015*). It has thus been proposed that the partitioning of *pmf* into $\Delta\psi$ and $\Delta pH$ is regulated to maintain a balance between efficient energy storage and regulation of light capture (*Cruz et al., 2005*). We demonstrate here another important constraint on this balance: the avoidance of photodamage caused by recombination reactions in PSII, and this may explain the need for complex ion balancing systems in chloroplasts (*Cruz et al., 2001*; *Armbruster et al., 2014*; *Kunz et al., 2014*).

The effect of $\Delta\psi$ on recombination and $^1O_2$ production may, at least in part, explain the severe effects of fluctuating light on photosynthesis, and thus could constitute a significant limitation to photosynthetic productivity. In the absence of a large $\Delta\psi$, the energy gap between the $P^+Pheo^-$ and $P^+Q_A^-$ appears to be sufficient to keep detrimental recombination to a manageable level when the usual regulatory mechanisms are functional (*Rutherford et al., 2012*). During photosynthesis, and especially under fluctuating light, though, the energy gaps between the photo-generated radical pairs vary dynamically under the influence of the *pmf*, so that high $\Delta\psi$ renders the charge-separated states in the photosystems considerably less stable. The $\Delta\psi$ is also expected to influence the trap depth (i.e. the energy level between $P^*$ and the first radical pair(s), the most relevant probably being $P^+Pheo^-$) and this could potentially affect the quantum yield of charge separation and the yield of radiative recombination (luminescence).

Transthylakoid electric fields also influence recombination reactions in PSI reaction centers (*van Gorkom, 1996*). It is thought that $P_{700}$ triplet formation is minimized in PSI by the presence of the higher potential quinone in PsaA, making this side of the reaction center the safe charge recombination pathway (*Rutherford et al., 2012*). In light of the present findings, it is worth considering whether a transiently large $\Delta\psi$ could make this protective mechanism less efficient, though this question has yet to be addressed experimentally.

Over evolutionary time scales, the $\Delta\psi$-effect may have constrained other bioenergetics features of photosynthesis. It has long been known that oxygenic photosynthesis is limited, in most organisms, to wavelength ranges shorter than about 700 nm (the 'red limit'), resulting in the loss of a large fraction of the light energy hitting the plant (*Blankenship et al., 2011*). (*Gust et al., 2008*) proposed that the red limit may be the result of certain limitations imposed in part by key biochemical properties of life (e.g. the properties of energy storage molecules NAD(P)H and ATP, the use of certain biochemical pathways, etc.) that evolved before the advent of photosynthesis. Milo (*Milo, 2009*) came to a different conclusion based on estimates of the theoretical wavelength dependence of energy conversion efficiency for plant photosynthesis, based on the Shockley and Queisser equation (*Shockley and Queisser, 1961*). The maximum efficiency was about 700 nm, similar to the red limit of oxygenic photosynthesis, suggesting that evolution has selected for photosynthetic energetics based on this fundamental limit, rather than any biological imperative. However, the predicted wavelength-dependence of the energy efficiency is very broad, with only about a 15% decrease from the peak at wavelengths out to 800 nm, far beyond the red limit, even in organisms with red-shifted reaction centers. Marosvolgyi and van Gorkom (*Marosvolgyi and van Gorkom, 2010*) drew a similar conclusion but with additional restraints and suggested a narrower maximum more closely overlapping with the red absorption of chlorophyll *a*.

Rutherford et al. (*Rutherford et al., 2012*) took a different view based on an analysis of the bioenergetics of reaction centers. They noted that PSII was in a uniquely difficult situation in energy terms: (1) it does multi-electron chemistry (at both sides of the reaction center) and so cannot prevent back reactions by kinetic control, (2) it has a very energy demanding reaction to do: water oxidation and quinone reduction with a $\Delta E$ ~920 meV in functional conditions, and requires a significant over-potential not just for attaining a high quantum yield of photochemistry but also for achieving water oxidation and quinol release, and (3) its chlorophyll cation chemistry is uniquely oxidizing and thus it cannot use carotenoids to protect itself from chlorophyll triplet formation at the heart of the reaction center. This situation means that unlike other type-II reaction centers, PSII is unable to prevent electrons from the bound semiquinones getting back to the Pheo and then recombining with $P^+$ and forming $^3P$.

This lack of energy 'headroom' was seen not only as a major factor in PS II's susceptibility to photodamage but also as a reason why oxygenic photosynthesis is pinned to chlorophyll *a* photochemistry at around 680 nm as the red limit. The existence of efficient oxygenic photosynthesis at longer wavelengths seemed to question that view (*Miyashita et al., 1996*; *Kuhl et al., 2005*; *Gan et al., 2014*; *Behrendt et al., 2015*). However, it was pointed out that these species seem to exist in very stable environments that have very little variation in light conditions (*Cotton et al., 2015*). Under such a narrow range of illumination conditions it is not unreasonable that less energy 'headroom' is required (*Cotton et al., 2015*).

Clearly, these specific energy limitations of PSII will be exacerbated by spikes in the $\Delta\psi$ reported here. Indeed, it seems reasonable to suggest that the existing 'energy headroom' postulated in normal chlorophyll *a*-containing PSII, while too small to avoid photodamage altogether, exists quite specifically to mitigate the extra photodamage from back-reactions enhanced by spikes in $\Delta\psi$ due to variable light intensities. In this way, the extent of the variable light-induced $\Delta\psi$ may be considered to contribute to the position of the red limit of oxygenic photosynthesis.

The need to prevent $\Delta\psi$-induced recombination may also have guided the evolution of other photosynthetic components. Recent mechanistic models of the ATP synthase suggest that the ratio of protons passed through the ATP synthase per ATP synthesized depends on the number of subunits in the *c*-ring (*Silverstein, 2014*). The chloroplasts of green plants and algae thus far studied possess ATP synthase complexes with larger *c*-ring stoichiometries than their mitochondrial and bacterial homologues (*Seelert et al., 2000*), imposing higher fluxes of protons to generate ATP and necessitating the engagement of additional bioenergetic processes, including cyclic electron flow, to make up the ATP deficit needed to sustain photosynthesis (*Kramer et al., 2004*; *Avenson et al., 2005*).

While this increased H$^+$ demand is often viewed as a bioenergetic limitation to photosynthetic electron and proton transfer, a high H$^+$/ATP ratio decreases the *pmf* needed to maintain a given ATP free energy state ($\Delta G_{ATP}$), thus allowing photosynthesis to operate at a decreased steady-state *pmf*. In this context, the deleterious electron recombination effects of a high *pmf* may have favored the evolution of ATP synthase complexes with high H$^+$/ATP ratios in chloroplasts.

## Materials and methods

### Plant materials and growth conditions

Wild type *Arabidopsis thaliana* (ecotype Wassilewskija-2) and ATP synthase γ-subunit mutants were germinated on Murashige and Skoog medium supplemented with 2% (w/v) sucrose, and 10 mg L$^{-1}$ sulfadiazine for selection of transgenic *minira* lines (*Hadi et al., 2002*). Following germination plants were grown on soil under a 16 hr photoperiod at 100 µmol photons m$^{-2}$ s$^{-1}$ at 22°C for three weeks.

*Nicotiana tabacum* wild type (cv Samsun NN) and *ATPC1* antisense lines were germinated and grown as in (*Rott et al., 2011*) under a 16 hr photoperiod at 300 µmol photons m$^{-2}$ s$^{-1}$. Measurements were performed at the onset of flowering on the youngest, fully expanded leaves.

### Generation of chloroplast ATP synthase γ-subunit *minira (mini*mum *r*ecapitulation of *ATPC2)* mutants

A T-DNA insertion mutant for ATPC1 (*dpa1*) (*Dal Bosco et al., 2004*) was used to introduce the mutated constructs as a complemented allele as in (*Kohzuma et al., 2012*). Site-directed mutations in the redox-regulatory domain of ATPC1 were designed to incorporate amino acid differences from the redox inactive ATPC2 into the redox regulated ATPC1 (*Figure 1A*). The *Arabidopsis thaliana* AtpC1 gene was excised from binary vector pSex001 as a SmaI/XbaI fragment and cloned into SmaI/XbaI digested pBluescript plasmid. The resulting plasmid was named pDA15. The mutations were introduced into ATPC1 using a combination of three approaches.

The first approach used an adaptor ligation strategy. Oligonucleotides were designed to introduce desired mutations. Adaptors representing the 5' and 3' strand of DNA targeting specific mutations were obtained independently from Sigma Aldrich. A total of 5 ml of each oligonucleotide pair (10 mM) were mixed and denatured at 95°C for 10 min in a boiling water bath. The oligonucleotides were allowed to reach room temperature over two hours in the water bath allowing for efficient annealing of complementary strands. The adaptors thus obtained were utilized for adaptor ligation.

The target region from ATPC1 cDNA was removed using BglII/HpaI restriction enzymes. The larger linearized backbone of pDA15 was used for ligation with the adaptors (*Supplementary file 1*). The resultant vector was double digested with SmaI and XbaI to excise the mutated ATPC1 gene, which was then ligated back into the SmaI/XbaI digested binary vector, pSex001.

A second set of mutations was introduced by first digesting pDA15 with BglII followed by a partial digestion with TatI. The resulting plasmid backbone was used for ligation with the adaptors having the desired mutation (*Supplementary file 1b*). As in the first adaptor ligation-mediated mutagenesis approach, the resultant vector was double digested with SmaI and XbaI to excise the respective mutated atpC1 gene and ligated into SmaI/XbaI digested binary vector, pSex001.

A second mutagenesis strategy used splicing by overlap extension (SOE) PCR. For each desired mutation, two sets of primers were designed to produce two overlapping fragments during amplification of atpC1 from pDA15 such that the mutation was generated in the region of overlap (*Supplementary file 1c*). The two fragments were mixed together and the resulting DNA solution was used as a template for a subsequent PCR using primers that amplify the complete AtpC1 gene (DMP 45 and DMP 46). This led to the amplification of the entire atpC1 gene with the desired mutation. The mutated atpC1 was digested with SmaI and XbaI and was sub-cloned into SmaI/XbaI digested pSex001.

The third mutagenesis approach involved swapping of target domains (delete swaps) using synthetic gene fragments synthesized at GenScript USA (*Supplementary file 1*). The synthetic gene was used to replace the ATPC1 redox regulatory domain in the native gene. To swap the domains, pDA15 was digested with BsrGI and XbaI to remove the native domain and ligated with the

synthetic fragment derived from the synthetic gene construct after digestion with BsrGI and XbaI. To introduce a synthetic gene with a single nucleotide mutation, the synthetic gene construct for *minira* 3 and pDA15 were double digested with BsrGI and XbaI and the synthetic BsrGI/XbaI fragment with the mutation was ligated into pDA15 where the original BsrGI/XbaI fragment had been removed The resultant intermediate plasmid was digested with SmaI/XbaI to obtain the gene with swapped C domain. This gene was then sub-cloned into SmaI/XbaI digested pSex001. All of the introduced mutations were confirmed by Sanger sequencing of individual plasmids.

Following successful mutagenesis, *minira* constructs were mobilized into the binary vector pSEX001-VS under control of the Cauliflower mosaic virus 35S promoter. A single *minira* construct was transformed into heterozygous *dpa1* plants via *Agrobacterium tumefaciens*-mediated transformation (*Clough and Bent, 1998*). The resulting transgenic plants were screened for the *minira* insertion via PCR using the forward primer 5' -GGTAATATCCGGAAACCTCC- 3' and the reverse primer 5' -GTACAAGAGCTCGACTTTCTCG- 3' followed by *dpa1* screening using the forward primer 5' -CACATCATCTCATTGATGCTTGG- 3' and the reverse primer 5' -GTACAAGAGCTCGACTTTGTCG-3'. Transgenic plants containing both *minira* and *dpa1* insertions were self-pollinated until plants were homozygous for both *dpa1* ($\Delta atpc1$) and the correct *minira* mutation, which was subsequently confirmed by sequencing.

## Isolation of tightly coupled chloroplasts and intact thylakoids

Chloroplasts were extracted from market spinach with modifications to the method described in Seigneurin-Berny et al. (*Seigneurin-Berny et al., 2008*). All centrifugation steps were carried out at 4°C and exposure to light was kept to a minimum. Briefly, approximately 20 g of spinach leaves were homogenized in a blender for 10 s with ice cold homogenization buffer of 50 mM HEPES (pH 7.6), 330 mM sorbitol, 5 mM $MgCl_2$, 2 mM EDTA and supplemented with 0.1% BSA for grinding. The homogenate was filtered through three layers of wetted Miracloth and one layer of wetted muslin followed by centrifugation at 4000 x *g* for 10 min. The pellet was resuspended in homogenization buffer and layered on top of a single step 80%–40% Percoll gradient. Intact chloroplasts were recovered after centrifugation for 20 min at 3000 x *g* in a swinging bucket rotor. The intact chloroplasts were diluted approximately 4-fold with homogenization buffer and centrifuged for 5 min at 4000 x*g*. The chloroplast pellet was resuspended in a minimal amount of homogenization buffer (<1 mL) and chlorophyll quantified in 80% acetone (*Porra et al., 1989*).

## Spectroscopic measurements

Near simultaneous chlorophyll fluorescence and electrochromic shift (ECS) measurements were performed on a custom made spectrophotometer (*Hall et al., 2013*). For *in vivo* spectroscopic measurements, following a 10 min dark acclimation the maximal PSII quantum efficiency, linear electron flow (LEF), energy-dependent exciton quenching ($q_E$), and photoinhibitory quenching ($q_I$) were estimated using saturation pulse chlorophyll *a* fluorescence as described previously (*Livingston et al., 2010*). The extent of the $q_I$ component of NPQ was determined following at least 10 min dark relaxation to eliminate the residual effects of $q_E$ type quenching. Red actinic illumination was used for all measurements to prevent incorrect assessment of chloroplast movement as $q_I$, as red light is ineffective in inducing chloroplast movements (*Cazzaniga et al., 2013*). A Stern-Volmer derivation of $q_E$ ($q_{E(SV)}$) was used to minimize the contribution of $q_I$ in the determination of $q_E$ (*Krause and Jahns, 2003*). Estimates of the relative redox status of $Q_A$ ($q_L$) were performed as described in (*Kramer et al., 2004*) after at least 10 min of actinic illumination. The relative extents of steady state *pmf* (ECS$_t$) and the conductivity of ATP synthase to protons ($g_H^+$) were measured using the dark interval relaxation kinetics of absorbance changes associated with the electrochromic shift (ECS) fit to a first-order exponential decay (*Sacksteder and Kramer, 2000*). Partitioning of the *pmf* was determined from deconvolution of the absorbance change at three wavelengths (505, 520 and 535 nm) around 520 nm during the dark interval ECS changes and the ECS steady-state ($\Delta\psi$) and ECS inverse ($\Delta$pH) were determined as in (*Takizawa et al., 2007*). Briefly, the total amplitude of the deconvoluted ECS signal following a rapid light/dark transition was used to estimate the total light-induced *pmf* (ECS$_t$). The steady-state $\Delta\psi$ component was determined from the extent to which the inverted ECS signal during the dark interval decreased from the steady-state baseline. The steady-state $\Delta$pH component was determined as the amplitude of the inverted ECS signal during the dark interval. The ECS

measurements were corrected for pigment variations by normalizing to chlorophyll content determined from acetone extraction as above. For tobacco measurements, the ECS measurements were normalized to the xenon-flash induced extent of the $\Delta A_{520\ nm}$ ECS rise. $P_{700}^+$ reduction kinetics were measured from the dark interval relaxation kinetics of the absorbance change at 810 nm after subtracting the 930 absorbance change (*Baker et al., 2007*).

*In vitro* chloroplast measurements were performed on a similar instrument described above modified to measure a cuvette held sample. Chloroplasts were osmotically shocked on ice in buffer containing 10 mM HEPES (pH 7.8) and 10 mM $MgCl_2$ to a final chlorophyll concentration of 20 µg ml$^{-1}$ supplemented with 5 µM spinach ferredoxin and 10 µM ascorbate. Where noted, thylakoids were treated with 50 µM decyl-ubiquinol to catalyze PSI cyclic electron transfer and generate a *pmf*, 50 µM 3-(3,4-dichlorophenyl)-1,1-dimethylurea (DCMU) to block PSII forward electron transfer, and 25 µM gramicidin to decouple the *pmf*. Fluorescence measurements were performed as above, with variable fluorescence measured after a 20 min dark adaptation after which the thylakoids were excited with a single 100 ms subsaturating actinic pulse. From a dark-adapted state, the application of a single turnover pulse will lead to rapid accumulation of $\Delta\psi$, due to the high buffering capacity of the thylakoid lumen as well as the low capacitance of the thylakoid membrane, allowing gramicidin to decouple $\Delta\psi$, as the $\Delta\psi$ primarily composes the *pmf* under these conditions (*Cruz et al., 2001*). The $F_0$ measurements for all samples were taken from the first measured point to avoid any actinic effects due to the measuring pulses themselves.

## Estimation of recombination rate

The $\Delta\psi$-induced enhancement of the rate of recombination from the $S_2Q_A^-$ state was estimated based on the change in the equilibrium constant for the sharing of electrons in the presence of $\Delta\psi$. Other states will also recombine (e.g. the $S_3Q_A^-$ state) but we use $S_2Q_A^-$ as a proxy because we expect most of these states to respond to changes in $Q_A$ and $\Delta\psi$ in similar ways. The rate of recombination from $S_2Q_A^-$ was calculated as:

$$v_r = [S_2Q_A^-] * k_r 10^{\frac{-\Delta E_{stab}}{0.06}} \tag{1}$$

where $[S_2Q_A^-]$ is the concentration of PSII centers with reduced $Q_A$, $k_r$ the intrinsic rate of recombination from $S_2Q_A^-$, and $\Delta E_{stab}$ is the free energy for stabilization of the charge separated state, expressed in eV. In the absence of a field and in the presence of DCMU where all $Q_A$ is reduced, $v_r$ is measured to be about 0.3 s$^{-1}$ (*Figure 5*), but in the uninhibited complex under steady state photosynthesis, this rate will be decreased proportionally by oxidation of $Q_A$, while $\Delta E_{stab}$ will be decreased by $\Delta\psi$ so that:

$$v_r = 0.3 * (1 - q_L) * 10^{\frac{-\Delta\varphi_{light-dark}}{60}} \tag{2}$$

where $1-q_L$ is an estimation of the fraction of $Q_A$ in the reduced form (*Kramer et al., 2004*), and $\Delta\psi_{light-dark}$ is the light-dark difference in electric field in mV. From Takizawa et al. (*Takizawa et al., 2007*) we obtained a factor for estimating $\Delta\psi$ from the extents of ECS, and correcting for the chlorophyll content as performed here, we obtain:

$$v_r = 0.3 * (1 - q_L) * 10^{\frac{-ECS_{ss}}{60}} \tag{3}$$

where $ECS_{ss}$ is the normalized ECS signal representing the light-dark difference in $\Delta\psi$ normalized to the chlorophyll content.

## Chlorophyll fluorescence imaging

*In vivo* whole plant chlorophyll *a* fluorescence imaging was performed in an imaging chamber equipped with 50W Bridgelux White LEDs (BXRA-56C5300, Bridgelux Inc., Livermore, California) for white actinic illumination (*Cruz et al., 2016*). Pre-illumination values for $F_0$ and $F_M$ were captured just prior to the beginning of the photoperiod and subsequent fluorescence parameters obtained using a 300 ms saturating actinic pulse at ~25,000 $\mu$mol photons m$^{-2}$ s$^{-1}$ and a Red LED matrix (Luxeon Rebel SMT High Power LED Red, LXM2-PD01-0050, Philips Lumiled, San Jose, California) to measure chlorophyll which was then captured by a CCD camera (AVT Manta 145 M) equipped with a near infrared long pass filter (RT-830, Hoya Glass). Plants were imaged over three consecutive 24 hr

photoperiods (*Figure 2A*, *videos 1–9*, timing and light intensities are described in *Supplementary file 2*). During the first day, the actinic light intensity remained constant at 100 $\mu$mol photons m$^{-2}$ s$^{-1}$ to collect growth chamber conditions. Days two and three represented ramped lighting perturbations. The photoperiod for day two was sinusoidal, beginning at 39 µmol photons m$^{-2}$ s$^{-1}$ and increasing in intensity by approximately 1.2 times every 30 min until midday where it peaked at 500 $\mu$mol photons m$^{-2}$ s$^{-1}$, after which the light intensity decreased at the same rate every 30 min. The photoperiod for day three was sinusoidal with brief fluctuations in light intensity. Starting at 39 $\mu$mol photons m$^{-2}$ s$^{-1}$, the light intensity was doubled after 15 min followed by 1.5-fold increase for 12 min and the cycle repeated until peak intensities of 1000 and 500 $\mu$mol photons m$^{-2}$ s$^{-1}$ were cycled through at midday, after which the sinusoidal fluctuations decreased at the same rate as the increases. Steady state values for NPQ parameters of chlorophyll fluorescence were captured prior to the ramp to the next light intensity on days two and three, or hourly on day one and calculated as noted above for the fluorescence spectroscopy. Sequences of images were captured with a 60 ms delay between images for a 15 frame total for each measurement pre- during and post-saturation flash followed by images taken to correct for artifacts due to residual electrons in the CCD array. Images were analyzed using open source software (ImageJ, NIH) modified in house to allow calculations of photosynthetic fluorescence parameters across selected regions of interest.

## Photoinhibition of detached leaves

Plant leaves were excised and incubated in the dark for 3 hr with their petioles submerged in either water or 3 mM lincomycin to inhibit chloroplast protein translation (*Tyystjarvi and Aro, 1996*). The leaves were then illuminated with red light at 1000 $\mu$mol photons m$^{-2}$ s$^{-1}$ using a red actinic light for indicated periods of time. During illumination, the petioles of the leaves remained submerged in the treatment solution. Following illumination, leaves were allowed to dark adapt for 20 min, after which the $F_0$ and $F_M$ values of chlorophyll fluorescence were measured in order to determine $F_V/F_M$. For tobacco plants, leaf discs were soaked in a lincomycin solution and fluorescence parameters determined at 600 $\mu$mol photons m$^{-2}$ s$^{-1}$.

To determine PSII activity, following photoinhibitory treatment, performed as above, leaves were dark adapted for 20 min and then vacuum infiltrated with a 50 µM DCMU solution. Analysis of PSII activity was determined from the amplitude of the $\Delta A_{520\ nm}$ ECS signal using two saturating single-turnover flashes provided by a xenon lamp spaced 200 ms apart. The amplitude of the second flash, corresponding to PSI centers capable of charge separation, was subtracted from the amplitude of the first flash, corresponding to both PSI and PSII centers capable of charge separation, to obtain the relative PSII photosystems capable of activity both before and after photoinhibition. Xenon flashes were judged to be fully saturating by ensuring that essentially identical results were obtained with a 50% weaker intensity.

## Protein analysis

Photoinhibited leaf samples were collected as described above and total leaf proteins were extracted as described in Livingston et al. (*Livingston et al., 2010*). Analysis of chloroplast ATP synthase complexes was carried out on 20 µg of protein using leaves collected from 3–4 plants taken from the growth chamber. Analysis of PsbA (D1) protein levels during photoinhibitory treatment was carried out on 30 µg of total protein from 3–4 leaves sampled during the photoinhibitory treatment time points as described above. Proteins were separated by SDS-PAGE and the ATPB (β-subunit) and PsbA (D1) proteins detected using commercially purchased antibodies (Agrisera, Vannas, Sweden).

## $^1O_2$ detection

Plant leaves were excised and incubated in the dark for 3 hr in 250 µM Singlet Oxygen Sensor Green (SOSG, Life Technologies) prepared according to manufacturer's instructions. Petioles were maintained below the liquid's surface during the infiltration and were wrapped in a Kimwipe soaked in water during imaging to prevent drying. For valinomycin treatments, leaves were vacuum infiltrated with either SOSG solution or with SOSG supplemented with 50 µM valinomycin and subsequently measured. Successful penetration of leaf epidermal and mesophyll cells, as well as SOSG penetrance

throughout the cells was confirmed via confocal microscopy. The leaves were imaged in a chamber equipped with the same lighting as above. Qualitatively similar data were obtained for *minira* 3–1 leaves under both white and red (650 nm) LED illumination, ensuring that photosensitization of SOSG was not responsible for the signals obtained (*Ragas et al., 2009*). Images were captured with a cooled CCD camera (AVT Bigeye G 132B-NIR) equipped with a 555 nm 10 nm band pass filter. Fluorescence excitation was provided via 458 nm LEDS (Cree Inc). Images were analyzed using ImageJ software.

### Data analysis

All spectroscopic data were analyzed and figures generated using Origin 9.0 software (Microcal Software). Statistical analyses of data were performed in R package, utilizing two-way ANOVA to test for significant effects on photoinhibition ($q_I$) from the interaction with either the $\Delta pH$ or $\Delta\psi$ component of the *pmf*.

## Acknowledgements

This work was supported by the U.S. Department of Energy (DOE), Office of Science, Basic Energy Sciences (BES) under Award number DE-FG02-91ER20021 and the MSU Center for Advanced Algal and Plant Phenotyping (CAAPP). AWR was supported by a Biotechnology and Biological Sciences Research Council (BBSRC) grant (BB/K002627/1) and the Royal Society Wolfson Research Merit Award.

## Additional information

### Competing interests

DMK: Founder and stock holder in Phenometrics, Inc. and a founder of the PhotosynQ.org project, these entities manufacture and distribute instrumentation for plant and algal phenotyping. However, neither of these technologies are used in the current work. The other authors declare that no competing interests exist.

### Funding

| Funder | Grant reference number | Author |
|---|---|---|
| Basic Energy Sciences | DE-FG02-91ER20021 | Geoffry A Davis<br>Atsuko Kanazawa<br>Kaori Kohzuma<br>John E Froehlich<br>Mio Satoh-Cruz<br>David M Kramer |
| Biotechnology and Biological Sciences Research Council | BB/K002627/1 | A William Rutherford |
| Royal Society | Wolfson Research Merit Award | A William Rutherford |
| Michigan State University Center for Advanced Algal and Plant Phenotyping | | David M Kramer |
| Michigan State University Ag-BioResearch | | David M Kramer |

The funders had no role in study design, data collection and interpretation, or the decision to submit the work for publication.

### Author contributions

GAD, AK, Conception and design, Acquisition of data, Analysis and interpretation of data, Drafting or revising the article; MAS, DMK, Conception and design, Analysis and interpretation of data, Drafting or revising the article; KK, AD, Conception and design, Drafting or revising the article; JEF, ST, Acquisition of data, Analysis and interpretation of data; AWR, Analysis and interpretation of data,

Drafting or revising the article; MS-C, Conception and design, Analysis and interpretation of data; DM, Conception and design, Acquisition of data, Drafting or revising the article

## Author ORCIDs

David M Kramer, http://orcid.org/0000-0003-2181-6888

## Additional files

### Supplementary files

• Supplementary file 1. Oligonucleotides used for site directed mutagenesis of atpc1. (a) Oligonucleotide sequences utilized for adapter ligation mutagenesis. (b) Oligonucleotide sequences utilized for adapter ligation mutagenesis to introduce secondary mutations. (c) Oligonucleotide sequences utilized for splicing by overlap extension PCR. (d) Synthetic gene constructs incorporating multiple ATPC2 mutations into ATPC1.

• Supplementary file 2. Timing and light intensity profiles used for chlorophyll fluorescence imaging. (a) Timing and light profile of imaging day one. (b) Timing and light profile of imaging day two. (c) Timing and light profile of imaging day three.

• Supplementary file 3. Chlorophyll content of wild-type (Ws-2) and *minira* leaves. All measurements were performed on three-week old leaves following *in vivo* spectroscopic measurements as described in Materials and methods. Data represent the mean ± s.d. for n ≥ 3 leaves. Statistically significant differences (*p<0.05) from wild-type were determined using a t-test.

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
