## [Decision Letter]

Thank you for submitting your article "Limitations to photosynthesis by proton motive force-induced photosystem II photodamage" for consideration by *eLife*. Your article has been reviewed by two peer reviewers, including Robert Burnap (Reviewer #2), and the evaluation has been overseen by Kris Niyogi as the Reviewing Editor and Ian Baldwin as the Senior Editor.

The reviewers have discussed the reviews with one another and the Reviewing Editor has drafted this decision to help you prepare a revised submission.

Summary:

This manuscript deals with mechanisms of photoinhibition of photosystem II in vivo, which are still unresolved despite decades of study. The authors use a series of *Arabidopsis "minira*" mutants affecting the chloroplast ATP synthase γ subunit to show an unexpected relationship between the electric field (*Δψ*) component of the *pmf* and q_I_ (as a measure of photodamage to PSII). These results suggest a novel mechanism of PSII photoinhibition that involves an effect on PSII charge recombination, which in turn might result in increased singlet oxygen production. The data showing a correlation between *Δψ* and q_I_ are compelling and convincing, but the presentation and discussion of the subsequent results could be improved.

Essential revisions:

1) Introduction, second paragraph: The authors note that the present results go against the previous ideas that lumen acidification (not *Δψ*) was a significant cause of photoinhibition. However, we are not aware of experiments demonstrating that lumen acidification is a cause of photoinhibition in leaves, although the supposition can be found frequently in the literature. Were there experiments that showed acidification as a cause in vivo? If so, the reference needs to be more clearly cited. If not, then references to the proposition or the circumstantial evidence are more appropriate.

2) Figure 1 needs a loading control to enable comparison of the expression levels of ATPC1 mutant proteins in different *minira* lines.

3) In Figure 3, the "decreased levels of D1 protein" mentioned in the text are not obvious and convincing from the single immunoblot shown in the figure. In addition to showing a representative blot, these data for lincomycin-treated leaves (and the water-treated control) should be plotted in a format like the other data in this figure, and the number of biological replicates should be indicated in the legend. These data are important to demonstrate that the increase in q_I_ is indeed related to D1 photodamage rather than a type of sustained non-photochemical quenching.

4) Concerning Figure 4, the authors state that this shows that there is no correlation between the extent of PSII damage (q_I_) and Q_A_ redox state (q_L_) – they show three sub-figures, one for each light intensity used. If taken together (i.e., if the data had been plotted in one figure), would the relationship between q_L_ and q_I_ have much worse than that in Figure 5, which shows the relationship between *Δψ* and q_I_? This should be shown. The coexistence of two damage mechanisms would not be a surprise or a problem, nor would the entanglement of Q_A_ redox state on the likelihood of voltage-driven backreaction-induced damage.

5) In Figure 6, the authors switch from *Arabidopsis* to spinach thylakoids, although the text and legend indicate that the ECS signal is similar to that seen in *Arabidopsis*. For consistency, why was this experiment not performed with *Arabidopsis* thylakoids? Also, what is the evidence that the *pmf* is "stored almost exclusively as *Δψ*"? (There is no gramicidin or valinomycin treatment shown in panel B.)

6) Figure 7—figure supplement 1 shows that SOSG does not appear to be in chloroplasts, so this raises a question about the origin of the singlet oxygen that is being detected in Figure 7. Given the very short lifetime of singlet oxygen, it seems unlikely that it could diffuse from a site of generation in the PSII reaction centers in thylakoids, through the stroma and across the double envelope membrane of the chloroplast to react with SOSG in the cytosol. Also, how quantitative are the measurements of singlet oxygen using SOSG? A very recent review of this topic (Koh and Fluhr 2016 Plant Signal Behav, in press subsequent to submission of this manuscript) indicates that one should be cautious about quantification using SOSG. Some discussion of the limitations of this probe seems warranted.

[Editors' note: further revisions were requested prior to acceptance, as described below.]

Thank you for resubmitting your work entitled "Limitations to photosynthesis by proton motive force-induced photosystem II photodamage" for further consideration at *eLife*. Your revised article has been favorably evaluated by Ian Baldwin as the Senior editor and a Reviewing editor.

The manuscript has been improved but there are some remaining issues that need to be addressed before acceptance, as outlined below:

1) A loading control for Figure 1 is still needed. We apologize for not being clear about this essential revision in the previous decision letter. What is needed is not a dilution series of the wild type, which was already included in the figure, but additional panels showing that each lane of the blots was loaded with an equal amount of protein. This control is necessary to compare the expression level of ATPB in the different *minira* lines. This can be done by probing the blots with an antibody recognizing an unrelated protein whose level is the same in all lines, or at a minimum by showing the Coomassie-stained gels (as was done in the original Figure 3) or Ponceau-stained blots to demonstrate equal protein loading of the lanes.

2) As requested, Figure 3 was changed to a graph based on biological replicates, however from the error bars it is not clear that the difference between *minira* 3-1 and the wild type is significant. Statistical analysis of these data is needed. Also, the authors might want to consider showing these data in a line graph instead of a bar graph, for consistency with the corresponding data in Figure 3.

3) The response to essential revision #6 in the previous decision letter was not completely clear. In their response, the authors state that "The localization of SOSG under a confocal microscope can generate SOSG fluorescence due to the intense laser light used to excite the dye, and therefore would not be useful in our understanding of physiological ^[1]^O_2_ generation." If this is the case, then Figure 7—figure supplement 1 should be omitted from the manuscript.

---

## [Author Response]

Essential revisions:

*1) Introduction, second paragraph: The authors note that the present results go against the previous ideas that lumen acidification (not Δψ) was a significant cause of photoinhibition. However, we are not aware of experiments demonstrating that lumen acidification is a cause of photoinhibition in leaves, although the supposition can be found frequently in the literature. Were there experiments that showed acidification as a cause in vivo? If so, the reference needs to be more clearly cited. If not, then references to the proposition or the circumstantial evidence are more appropriate.*

Lumen acidification-induced PSII inactivation or photodamage was proposed earlier based on in vitro data (Krieger and Weis, 1993 Photosynth Res 37, 117-130, Johnson et al. 1995, Biochim biophys acta 1229, 202-207, Kramer et al., 1999 Photosynthesis Res 60, 151-163) but was not directly demonstrated in vivo. Indeed, testing this hypothesis was one of the motivations for the current study. We have clarified this point in the text (see Introduction, second paragraph).

*2) Figure 1 needs a loading control to enable comparison of the expression levels of ATPC1 mutant proteins in different minira lines.*

Figure 1 contains a loading control, outlined and labeled as Ws-2, with the corresponding values of 100%, 50%, and 25% shown above the bands for comparison of wild-type (Ws-2) relative content with the *minira* mutants. As the number of mutant lines analyzed in Figure 1 required the use of multiple gels, loading controls were run on each gel to ensure that differences in development between gels would not be a factor.

*3) In Figure 3, the "decreased levels of D1 protein" mentioned in the text are not obvious and convincing from the single immunoblot shown in the figure. In addition to showing a representative blot, these data for lincomycin-treated leaves (and the water-treated control) should be plotted in a format like the other data in this figure, and the number of biological replicates should be indicated in the legend. These data are important to demonstrate that the increase in q_I_ is indeed related to D1 photodamage rather than a type of sustained non-photochemical quenching.*

We have updated Figure 3 to include a graph of quantified D1 protein levels from multiple biological experiments of leaves treated with lincomycin. While differences between individual leaves and western blots show some variation, the *minira* 3-1 D1 protein levels decrease faster than wild-type leaves under the same conditions as the other data presented in Figure 3, confirming that the photoinhibition phenotypes observed correspond to damage to PSII rather than sustained PSII quenching.

*4) Concerning Figure 4, the authors state that this shows that there is no correlation between the extent of PSII damage (q_I_) and Q_A_ redox state (q_L_) – they show three sub-figures, one for each light intensity used. If taken together (i.e., if the data had been plotted in one figure), would the relationship between q_L_ and q_I_ have much worse than that in Figure 5, which shows the relationship between Δψ and q_I_? This should be shown. The coexistence of two damage mechanisms would not be a surprise or a problem, nor would the entanglement of Q_A_ redox state on the likelihood of voltage-driven backreaction-induced damage.*

We appreciate the reviewer’s thoughtful comment on this data and have taken the opportunity to expand our discussion about the potential influence of the Q_A_ redox state on photoinhibition and recombination rates.

First, in preparing the manuscript, we plotted the data in the three panel view to allow the readers to disentangle light intensity and mutation-induced effects, but now present the data in one panel (see Figure 4). There is, indeed, a negative relationship between q_L_ and q_I_, but most of this is dependent on light and not on the mutation. In other words, at each light intensity, we observed large dependence of q_I_ between mutant lines even though q_L_ remained fairly constant (see Figure 4) making the case that Q_A_ redox state cannot by itself explain the effects.

However, we completely agree with the reviewer that the rate of recombination should be dependent not only on △*ψ*, but also on Q_A_ redox state, and that if we consider both we should be able to see the expected relationship even across different light intensities. To test for this, we derived an equation (see subsection “Estimation of recombination rate” and Figure 5) to estimate the combined effects of Q_A_ redox state and △*ψ* on recombination through P_680_Pheo^-^. As shown in Figure 5, panel C, we do see a positive relationship between photoinhibition and this predicted recombination rate, though there is some scatter particularly with the wild type that suggests (not surprisingly) there are other factors that also influence photoinhibition, as discussed in the revised text. Overall, we are very happy that the reviewer suggested this point because it allowed us to present a more mechanistic view that supports the overall hypothesis.

*5) In Figure 6, the authors switch from Arabidopsis to spinach thylakoids, although the text and legend indicate that the ECS signal is similar to that seen in Arabidopsis. For consistency, why was this experiment not performed with Arabidopsis thylakoids? Also, what is the evidence that the pmf is "stored almost exclusively as Δψ"? (There is no gramicidin or valinomycin treatment shown in panel B.)*

Unfortunately, thylakoids from *Arabidopsis* tend to be very leaky to protons so that the *pmf*-induced effects are negated, and this effect can be quite variable from preparation to preparation. This is why we chose spinach, a well-established model for this type of experiment, as intact thylakoids are easily isolated.

As discussed within the text (Results, eighteenth paragraph), during the rapid transition from a dark adapted state to the light, the buffering capacity of the thylakoid lumen leads to a lag in the formation of a pH gradient, resulting in an immediate *pmf* formation that is composed almost exclusively as △*ψ*.

*6) Figure 7—figure supplement 1 shows that SOSG does not appear to be in chloroplasts, so this raises a question about the origin of the singlet oxygen that is being detected in Figure 7. Given the very short lifetime of singlet oxygen, it seems unlikely that it could diffuse from a site of generation in the PSII reaction centers in thylakoids, through the stroma and across the double envelope membrane of the chloroplast to react with SOSG in the cytosol. Also, how quantitative are the measurements of singlet oxygen using SOSG? A very recent review of this topic (Koh and Fluhr 2016 Plant Signal Behav, in press subsequent to submission of this manuscript) indicates that one should be cautious about quantification using SOSG. Some discussion of the limitations of this probe seems warranted.*

We agree that one should be careful in making quantitative estimates of ROS based on dyes like SOSG, and have restricted our discussion to relative increases within our different treatments. We explicitly point out these caveats in the text (Results, fourteenth paragraph and subsection “^[1]^O_2_ detection”).

Regarding the location of singlet oxygen generation in our samples, while we cannot specifically localize SOSG to the chloroplast, our confocal images imply that is it not excluded from thylakoid, as has been speculated by other groups (Hideg 2008, Central European J Biol 3(3), 273-284). However, it must be kept in mind that the SOSG fluorescence observed in Figure 7 and Figure 7—figure supplement 2 were observed under specific light conditions (described within the figure legend). The localization of SOSG under a confocal microscope can generate SOSG fluorescence due to the intense laser light used to excite the dye, and therefore would not be useful in our understanding of physiological ^[1]^O_2_ generation.

While dye penetration cannot be targeted, and is clearly present throughout the cell and leaf, the increases we observed using SOSG strongly agree with our model that the singlet oxygen is produced as a function of photosynthetic reactions. The relationship between PSII recombination reactions, electric field effects on these reactions, as well as changes in SOSG fluorescence when the electric field is altered with valinomycin are all strong indicators that the SOSG signals are due to photosynthetic reactions.

[Editors' note: further revisions were requested prior to acceptance, as described below.]

*1) A loading control for Figure 1 is still needed. We apologize for not being clear about this essential revision in the previous decision letter. What is needed is not a dilution series of the wild type, which was already included in the figure, but additional panels showing that each lane of the blots was loaded with an equal amount of protein. This control is necessary to compare the expression level of ATPB in the different minira lines. This can be done by probing the blots with an antibody recognizing an unrelated protein whose level is the same in all lines, or at a minimum by showing the Coomassie-stained gels (as was done in the original Figure 3) or Ponceau-stained blots to demonstrate equal protein loading of the lanes.*

We apologize for the confusion about the loading control in Figure 1. We have incorporated a Coomassie Brilliant Blue stain of identically run samples as the ones used for the blots to show equal protein loading of the mutant samples to the 100% wild type sample.

*2) As requested, Figure 3 was changed to a graph based on biological replicates, however from the error bars it is not clear that the difference between minira 3-1 and the wild type is significant. Statistical analysis of these data is needed. Also, the authors might want to consider showing these data in a line graph instead of a bar graph, for consistency with the corresponding data in Figure 3.*

We appreciate the suggestions for improving Figure 3. We have incorporated another experiment into the figure (n=4), which improved the deviation between different samples. Over the course of the experiment displayed in Figure 3, the change in D1 content from the initial values is statistically significant between the wild type and *minira* 3-1, confirming that the q_I_ quenching that we measured does not represent a long-lived quenching state but indeed a loss of PSII. We have also changed the representation for Figure 3 to a line graph for consistency.

*3) The response to essential revision #6 in the previous decision letter was not completely clear. In their response, the authors state that "The localization of SOSG under a confocal microscope can generate SOSG fluorescence due to the intense laser light used to excite the dye, and therefore would not be useful in our understanding of physiological ^1^O_2_ generation." If this is the case, then Figure 7—figure supplement 1 should be omitted from the manuscript.*

We performed the confocal experiment to address test the possibility that SOSG does not adequately penetrate leaf cell layers (Hideg, E (2008) Central European Journal of Biology3(3)273-284), i.e. to convince ourselves that the method we employed was successfully delivering the SOSG dye throughout the leaf. The confocal microscopy images confirm that SOSG does indeed penetrate the cell layers and in this respect is a useful control. (We indicated that the confocal laser can induced SOSG photochemistry only to clarify that the images themselves do not represent strong background levels of ^1^O_2_ production). However, others groups (Dall’Osto et al. 2012, Ramel et al. 2013, Shumbe et al. 2016, cited at lines 263-246) have also presented evidence that SOSG adequately penetrates the cells of intact leaves, so our results are only confirmatory so we have removed the figure and associated discussion from the manuscript.